# Evidence for Alternate Stable States in an Ecuadorian Andean Cloud Forest

Ana Mariscal [1,2], Daniel Churchill Thomas [3,*] , Austin Haffenden [4], Rocío Manobanda [2], William Defas [2], Miguel Angel Chinchero [2], José Danilo Simba Larco [2], Edison Jaramillo [2], Bitty A. Roy [5] and Mika Peck [6]

1 Cambugan Foundation, Quito 170521, Ecuador; amariscal2005@yahoo.com
2 Instituto Nacional de Biodiversidad, Herbario Nacional del Ecuador (QCNE), Quito 170529, Ecuador; rociomanobanda@yahoo.com (R.M.); williandefas_m@yahoo.com (W.D.); cdtk@hotmail.com (M.A.C.); jdlarco2@hotmail.com (J.D.S.L.); bioedijara78@hotmail.com (E.J.)
3 BayCEER Institute, 95440 Bayreuth, Germany
4 School of Environment and Natural Resources, University of Wales Bangor, Gwynedd LL57 2UW, UK; aushaff@zoho.eu
5 Institute of Ecology and Evolution, University of Oregon, Eugene, OR 97403, USA; bit@uoregon.edu
6 Ecology, Behaviour and Environment, School of Life Sciences, University of Sussex, Brighton BN1 9RH, UK; m.r.peck@sussex.ac.uk
* Correspondence: daniel.thomas@uni-bayreuth.de

**Abstract:** Tree diversity inventories were undertaken. The goal of this study was to understand changes in tree community dynamics that may result from common anthropogenic disturbances at the Reserva Los Cedros, a tropical montane cloud forest reserve in northern Andean Ecuador. The reserve shows extremely high alpha and beta tree diversity. We found that all primary forest sites, regardless of age of natural gaps, are quite ecologically resilient, appearing to return to a primary-forest-type community of trees following gap formation. In contrast, forests regenerating from anthropogenic disturbance appear to have multiple possible ecological states. Where anthropogenic disturbance was intense, novel tree communities appear to be assembling, with no indication of return to a primary forest state. Even in ancient primary forests, new forest types may be forming, as we found that seedling community composition did not resemble adult tree communities. We also suggest small watersheds as a useful basic spatial unit for understanding biodiversity patterns in the tropical Andes that confound more traditional Euclidean distance as a basic proxy of dissimilarity. Finally, we highlight the conservation value of Reserva Los Cedros, which has managed to reverse deforestation within its boundaries despite a general trend of extensive deforestation in the surrounding region, to protect a large, contiguous area of highly endangered Andean primary cloud forest.

**Keywords:** Reserva Los Cedros; tropical forest ecology; tropical Andean biodiversity hotspot; alternate stable states; tropical forest conservation; forest succession

## 1. Introduction

Succession has been a central topic of discussion in ecology for at least a century [1–11]. As with many emergent properties of ecosystems, succession in ecosystems is endlessly controversial: the very existence of stable equilibria states and predictable successional (seral) stages have long been both called into question and defended [2,9,12,13]. In recent times, increasing attention has been given to models that lie within a community assembly and/or coexistence framework to explain and predict community composition in plants [14,15]. Rather than attempting to broadly predict changes in dominant species associations in the form of seral stages, community-assembly and coexistence models give greater attention to the importance of individuals, which sum to explain differences in community compositions. These models emphasize dispersal, priority, and stochastic effects on individuals, and in the case of niche-based or trait-based models, a further discussion of selection or "filtering" by biotic and abiotic factors [16–21]. There is no inherent contradiction among these

two broad approaches to community modeling: successional and assembly/coexistence approaches are highly complementary [15,22], and are sometimes packaged into a general "dynamics" framework [23]. This cultural shift in community ecology is perhaps due to a desire by ecologists to move away from the highly idiosyncratic, localized knowledge of a site that is often required to successfully predict successional patterns [12,24], and instead use more abstract and more universal ecological models.

This shift may also perhaps be partly due to the chaotic times in which we find ourselves. In the current era, most ecosystem types are experiencing historical rates of species loss [25,26], and are already undergoing consequences of climate change [27–33]. Ecosystems are also continually receiving new species from human activity [34] and face direct modification or even wholesale elimination due to land-use change [35–37]. Few anthropogenically disturbed ecosystems seem to have been successfully "returned" to stable historical ecological states [34], let alone to states that resemble some type of prehistoric ecological stability. In our destabilized era, the search for ancient species associations that require multiple uninterrupted, decadal journeys through intermediate ecological stages can sometimes seem quixotic.

However, the discussion of succession in forests has taken on particular and new urgency in the current era of accelerating climate change. Forests and their soils fix a significant portion of carbon every year, which has probably been essential in slowing negative effects of anthropogenic atmospheric carbon release [38]. Forests and their soils are also sitting reservoirs of vast amounts of sequestered carbon in soils that are readily lost through disturbance [39]. After many decades of underappreciation as essentially carbon-neutral ecosystems [40], a vibrant discussion has arisen around the role of primary forests not only as immense storehouses of carbon, but also as continuing carbon sinks [41,42]. New, proforestation-centered prescriptions for climate mitigation have therefore gained support: primary or late-stage successional forests should be protected as reserves of existing fixed carbon and potentially high-functioning sinks for further carbon fixation, and management of secondary forests should enhance characteristics that maximize carbon sequestration [43–46]. Such characteristics are often exemplified by old forests [47,48]. Tropical forests are the most volatile portion of the global forest carbon sink, and their role as a sink for atmospheric $CO_2$, rather than a source, will depend greatly on how they are managed and protected in the coming years [49]. Tropical montane forests are likely more significant carbon stores than was historically thought, given higher-than-predicted productivity and their large surface areas, resulting from complex, high-angle topography [50,51]. Thus, predictive models of succession may now become essential tools in mitigating the climate crisis: an understanding of seral stages (if indeed they exist!) in forests will be necessary for reaching maximal carbon storage and protecting existing carbon reserves.

The current study was undertaken at Reserva Los Cedros, a midelevation cloud forest reserve located in the heart of the tropical Andean biodiversity hotspot [52]. While the Tropical Andes may not display the same levels of alpha tree diversity often seen in the forests of the lower Amazon basin [53,54], the Tropical Andes have long been recognized for high biodiversity across multiple taxonomic groups [55–63]. This biodiversity is characterized by endemism at very fine spatial scales [64,65], making it an extremely important conservation priority [66]. The mechanisms for the high biodiversity and endemism in the tropics at large, including montane forests, have been debated for decades and perhaps centuries [67,68]. High solar energy and constant, plentiful rainfall are characteristic in many areas of the neotropics, and these traits are often associated with high biodiversity [69]. Additionally, the neotropics may have higher speciation rates, acting as evolutionary "cradles", and/or have higher retention of taxa, thus acting as refugia or "museums", allowing for greater accumulation of species over geological time [70,71]. The tropical Andes augment these general biodiversity trends in the tropics with characteristics that promote the role of endemism even further: complex topography creates diversified habitats and environmental gradients, as well as increased niches [58,67,72], increasing the potential

for sympatric speciation, and also adding dispersal barriers to increase the potential for fine-scale allopatric speciation [73]. The insular or "island" nature of cloud/mountain systems may be particularly enhanced situations for speciation events that result from neutral events, especially founder effects [72]. All of these factors can interact and sum to create the unique patterns of diversity and endemism observed in the Andes.

Here, we examined the dynamics of early (<20 years) forest succession at Reserva Los Cedros following anthropogenic or natural-gap-forming-disturbances. The landscape of the reserve is dominated by primary forest, but contains mixed-age secondary forests of varying land-use histories (see the Results section). In regions where remnants of ancient forest ecosystems—ecosystems without a history of significant modern anthropogenic disturbance—persist on the landscape, there are advantages for both ecological analysis and restoration efforts. First, we have information in the form of local, functioning examples of the biological complexity and reference points for primary-forest equilibria. Second, forest recovery is facilitated by the presence of species that have coevolved under local conditions for many millennia, available to act as nuclei for reforestation efforts [74,75]. The existence of extensive primary forest has proven particularly valuable for the study of succession and community equilibria in Neotropical forests [22,24,76].

We also examined the spatial signature of this fine-scale beta diversity of trees, with the working hypothesis that individual, steep-sided catchments may be the spatial unit of importance for understanding Andean biodiversity. We hypothesized that two characteristics/processes govern the behavior of community similarity in the tropical Andes: (1) large-scale spatial auto-correlation will cause short-distance comparisons to be more similar than farther comparisons, and with distance, all site comparisons will approach complete dissimilarity, a phenomenon known as Tobler's Law [77]; however, (2) we predicted significant "noise" around this general trend of decreasing similarity, because the complex topography of the Andes causes great dissimilarity even among some very localized comparisons, and conversely, causes highly similar habitats to occur sometimes at great distances apart due to similar site conditions. We predicted that in systems with extremely complex topography such as the Andes, the most informative unit for modeling community dissimilarity would not be Euclidean distance (meters), but instead the number of watersheds crossed. If this hypothesis was supported, it would facilitate future understanding of tropical diversity.

Finally, Los Cedros has succeeded at both forest protection and reforestation, despite its location in a region of high deforestation and now mineral exploration. Thus, we examined here the success of Los Cedros in the conservation of a forest in a region of Ecuador that is under intense extractivist pressure, in terms of forest cover change and IUCN red-list species observed.

## 2. Methods

### 2.1. Site—Reserva Los Cedros

All fieldwork described was performed at or directly adjacent to Reserva Los Cedros (www.reservaloscedros.org, accessed on 31 April 2022), a protected forest reserve on the western slope of the Andes, in northwestern Ecuador (00°18′031.000 N, 78°46′044.6′00 W), at 1000–2700 m asl. The reserve lies within the Andean Choco bioregion, one of the most biodiverse habitats on the planet [72,78,79]. It is also considered to be within the tropical Andean biodiversity hotspot [66]. The reserve protects 5256 hectares of cloud forest. Definitions of cloud forest vary and can be quite complex [80], but following Foster [81] and Stadtmüller [80], we use the term cloud forest to mean a forest "whose characteristics are tied to the frequent presence of clouds and mist", and consider the terms montane rain forest and cloud forest to be synonyms. Following Grubb et al. [82,83], the ecotones within the reserve are probably most accurately classified as a cloud forest of mostly lower montane rain forest, with some regions of higher montane cloud forest, and some Elfin forest zones in its highest, least-explored areas. The Reserve also shares a border with the 305,000 hectare nationally protected Cotocachi-Cayapas National Park. Rainfall

averages 2903 (±186 mm) per year according to onsite measurements. Humidity is typically high (~100%), and daily temperatures at the site range from 15 °C to 25 °C [84]. Annual fluctuations in temperatures are minimal. Daily precipitation varies according to the time of year, with the wet season (October–May) to dry (June–September) seasons [85].

The rarity of primary forest in the north Andean region is due to deforestation from precolonial times [76,86], and more recently in the 1960s due to land reform efforts that legalized and encouraged homestead-scale settlement of large government and private ("Hacienda") forest holdings [87]. Los Cedros has apparently largely escaped deforestation during both eras. The land within Los Cedros and the surrounding region was inhabited by the poorly understood Yumbo indigenous group until 1690. Their activities probably significantly altered sites, but with unknown effects on the ecology and canopy cover of the forest, and their economy was likely integrated into the forested setting of the midelevation Andean region [88,89]. However, the impacts of indigenous land use in South American forests has been, until recently, greatly underestimated and misunderstood by scholars [76,90]. Given the rainfall and high humidity, it is unlikely that large-scale deforestation resulted from fires at Los Cedros. Additionally, fire scares were not observed in the vicinity of the study, neither within nor outside the study plots. Other than possible small-scale precolonial indigenous activities, the majority of the Los Cedros forest has seen relatively few anthropogenic alterations.

Land acquisitions to build the reserve were made between 1988 and 1995. Records of ownership and land use prior to the establishment of the reserve are generally not available or are not highly reliable, so oral histories of anthropogenically altered sites were collected from Los Cedros staff and the community. Tracts purchased by Los Cedros were usually developed or deforested only in small proportions of their total area, usually for small-scale mixed ("finca") agriculture or for cattle and mule pasture. In some cases, these small-scale clearings were made for assertion of legal rights over land, rather than agricultural production at scale (Jose DeCoux, pers. comm.). Though acquisitions were made during the period of 1988–1995, use of these sites by previous landowners often apparently continued for years. Prior to abandonment to forest regeneration, sites of intensive agricultural use are understood to usually have been in use for longer periods of time compared to the pastured sites, as agricultural sites were often the sites of small homesteads or fincas. In contrast, pasture sites were often cleared merely to establish ownership before sale, or for temporary hosting of cattle herds.

Once under active management by Reserva Los Cedros, all sites began regeneration of the forest at approximately the same time. The approximate time of abandonment of agriculture or grazing for all sites was 6–7 years prior to the initiation of the survey. All former agricultural and pasture sites were selectively grazed by cattle to reduce competition from graminoids for protected tree seedlings. Tree seedlings were not planted, but were allowed to re-establish by encroachment from the adjacent forest during and after selective grazing. This method is known informally by some workers in the region as "reforestation by cattle rotation", and is intended to release naturally regenerated tree seedlings from intensive competition from pasture grasses [91] without the use of herbicides.

Size and growth forms were used to estimate tree age to corroborate oral history of each area of anthropogenic alteration. However, given the informal/incomplete historical record that was available, it was not possible to recover the exact length of human settlement prior to abandonment for most areas. Sites of plots were selected to be comparable among their qualitatively classified use history (see the Survey Methods subsection below).

Satellite data on forest cover from 1990 shows ~96% forest cover of Los Cedros (see results), of which at least 80% is thought to have been primary forest. Non-forest land use was concentrated in the southeastern portion of the reserve, where the vegetation surveys were undertaken.

## 2.2. Tree Survey and Plant Identification

### 2.2.1. Selection of Sites and Categorization of Land-Use History

Primary forest was defined as those forests which had no historical record and no physical evidence of canopy removal or other large structural alteration by humans. However, forest ecosystems exist continuously as matrices of various states of nonanthropogenic gap formation and closure [3,92–96], including montane wet forests and cloud forests (Crausbay and Martin, 2016). As such, primary forest was divided into three categories of land cover based on their gap characteristics: Bosque Cerrado (Closed Forest) "BC" = mature forest with no physical signs of a gap-forming disturbance. Canopy is closed. Bosque Secundario (Secondary Forest) "BS" = sites with evidence of a recent gap, now with a closed canopy. Claros del Bosque (Forest Clearings) "CLB" = recent, natural gaps in the forest.

In addition to primary forest, sites with histories of anthropogenic disturbance were placed into two categories: Regeneración de Fincas Agricultura y Ganadería (Regeneration from Agriculture and Pasture) "RG" = abandoned small family farms, with land use mostly consisting of pasture maintained for cattle. Regeneración Caña de Azúcar (Regeneration from Sugar Cane) "RCA" = intensively farmed sites used for sugar cane or corn production.

### 2.2.2. Survey Methods

Site-selection surveys were undertaken at Reserva Los Cedros for the years of 2005 and 2007. The southern area of Los Cedros was divided into 4 areas of study, which were then further divided into three sub-blocks each, from which one sub-block was randomly selected and searched for natural gaps. Once located, each natural gap was also accompanied by a BS and BC site, at a minimum of 40 m distance between survey sites (Supplementary Materials Figure S1). Two additional smaller blocks were added in the vicinity of previously settled areas to increase coverage of anthropogenic disturbances. Overall, 61 sites from various land-use histories and elevations were sampled (see results.).

Adult trees were defined as trees at the height 1.5 m with a diameter of 10 cm or greater. Material from all trees fitting this description were sampled within a circle plot of 30 m$^2$ radius from the center point for morphological identification of species where possible. Additionally, within each survey site of BC, RCA, and RG plots, all juvenile trees were examined. Due to time constraints, juvenile trees from BS and CLR plots were not sampled. Juvenile trees were defined as trees with a maximum diameter less than 10 cm, growing from 50 cm up to 2 m of species known to be capable of growing to adult tree size as given above, under ideal circumstances and with sufficient time. Juvenile trees were surveyed within a square subplot of size 5 × 5 m that was centered within the larger 30 m circular plot. Nomenclatures of identifications were based on the *Flora of Ecuador* [97].

## 2.3. Statistical and Informatic Methods

All analyses were conducted in Python and R. Python version 3.8.10 [98], using Pandas 1.1.3 [99,100] and Matplotlib 3.1.2 [101] for visualization; and R version 3.6.3 [102] with the in-box R plotter engine were used. All analyses were conducted in an Ubuntu 20.04.2 LTS environment. Where Bayesian analyses were used, Python version 3.7.3 was used for compatibility with the PyMC3 package version 3.8 [103]. All Bayesian models used the default NUTS sampler to sample the posterior. The code for all pertinent statistical analyses was run and recorded using a Jupyter Notebook that was stored in the affiliated GitHub repo, and is viewable as a notebook online (https://nbviewer.jupyter.org/github/danchurch/losCedrosTrees/blob/master/anaData/MariscalDataExploration.ipynb, accessed on 31 April 2022).

### 2.3.1. Species Accumulation Curves and Richness Estimators

Species accumulation curves for the entire area studied were calculated as the number of adult tree species observed per meter-squared of the physically sampled area. Each subplot covered a circle with a diameter of 30 m, or 0.071 ha. Additionally, a permanent tree diversity plot was established concurrently with rapid surveys in 2005. Within this,

trees were sampled in a grid format, every 10 m along an east/west axis and every 5 m along a north/south axis, in a rectangular half-hectare area (50 × 100 m). Trees were identified as species where possible, and unidentified trees were grouped into species-level operational taxonomic designations (Supplementary Table S1). In the case of the permanent plot, tree species accumulation was modeled as a function of trees examined. Species accumulation models and species estimators were calculated using the specaccum, specpool, and associated functions with the Vegan package in R version 2.5–6 [104]. Point predictions of diversity for a 1 ha area were generated using the predict function, which when used as a method of specpool model objects, used a Mao Tau rarefaction method [105] to generate predictions.

2.3.2. Tree Community Turnover (Distance Decay or Beta Diversity)

Turnover in tree communities was modeled and visualized using the Bray–Curtis dissimilarity [1,106] as a function of (1) physical distance or (2) watershed crossings mapped at a small scale. The Bray–Curtis dissimilarities and physical distance matrices among subplots were calculated using the SciPy spatial submodule [107]. Bayesian models of community dissimilarity decay were conducted using PyMC3 package in Python, with visualizations created using the companion ArViz package [108].

Turnover by Physical Distance

Tree community turnover by physical distance was calculated using the Bray–Curtis dissimilarity as a function of physical distance (meters). Two Bayesian models were used as alternative formulations of the above two processes of local variability and large-scale spatial autocorrelation.

Asymptotic Model

In one approach, the Bray–Curtis dissimilarity was modeled as an asymptotic function, or "Michaelis–Menten"-type function:

$$y = \frac{x}{K_{\mathrm{m}} + x} \tag{1}$$

where $y$ is the predicted Bray–Curtis dissimilarity, $x$ is the distance between the compared sites, and $K_{\mathrm{m}}$ is the distance at which half of the maximum Bray–Curtis dissimilarity is reached. This model honors both theoretical constraints of complete similarity at proximal sites and maximum dissimilarity. Additionally, we modeled variation around the mean as a linear variable, reducing as all distant comparisons approached complete dissimilarity (Bray–Curtis dissimilarity = 1). This shrinking variance term $\epsilon$ was exponentiated to avoid negative values. As a Bayesian model, this was formulated in probabilistic terms, with priors, as follows:

$$y \sim \mathrm{N}\left(\frac{x}{K_{\mathrm{m}} + x}, \varepsilon\right), \varepsilon = e^{(\delta x + \gamma)} \tag{2}$$

$$K_{\mathrm{m}} \sim \mathrm{N}(\mu = 200, \sigma = 10) \tag{3}$$

$$\delta \sim \mathrm{N}(\mu = -0.0006, \sigma = 0.005) \tag{4}$$

$$\gamma \sim \mathrm{N}(\mu = 1.5, \sigma = 0.5) \tag{5}$$

Prior distribution for $K_{\mathrm{m}}$ was loosely estimated based on Draper et al. [53], who showed that in many Amazonian forest ecosystem types, the decay of tree community similarity occurs rapidly, losing more than half their similarity within 200 m. Owing to the novel formulation, priors for delta and gamma of Equation (2) were not found in the existing literature, and were assigned weak priors, with means intended to reflect the initial wide variation and its subsequent tightening around the Bray–Curtis dissimilarity = 1.

Linear Model

In a second formulation, the decay in tree community similarity was modeled as a simple linear equation. Variance around the mean Bray–Curtis dissimilarity was still allowed to vary negatively with distance, beginning with a large initial spread to encompass both highly similar and dissimilar sites at close distances. Additionally, a skewed mean distribution [109,110] was used to describe the variance around the mean function, given the long skew that was observable in the data at most distances. As with the asymptotic model, the variance around the mean function of the Bray–Curtis dissimilarity was described as an exponentiated linear function.

$$y \sim skewNormal(\alpha + \beta x, \varepsilon, \alpha), \ \varepsilon = e^{(\delta x + \gamma)} \tag{6}$$

$$A \sim N(\mu = 0.9, \sigma = 0.2) \tag{7}$$

$$B \sim N(\mu = 0.0001, \sigma = 0.00001) \tag{8}$$

$$A \sim N(\mu = -2.0, \sigma = 0.5) \tag{9}$$

$$\Delta \sim N(\mu = -0.0006, \sigma = 0.005) \tag{10}$$

$$\Gamma \sim N(\mu = -1.5, \sigma = 0.5) \tag{11}$$

Prior distribution for $\beta$ was again loosely estimated based on Draper et al. [53], using data from community turnover of Amazonian terra firme forests. All other priors were assigned as weakly informative priors.

Model Comparison

Comparisons of performance between these two models were conducted via posterior predictive checks on the data and variance explained (Bayesian $R^2$) using the sample_posterior_predictive command in the PyMC3 package and the r2_score command in the ArViz package. Bayesian $p$-values to indicate a balanced performance were calculated using the distribution of residual differences between the model predictions and observed data. To confirm Bayesian linear model results, a classic linear least-squares regression was also applied to the data, using a Wald test with a null hypothesis that the slope of community turnover was zero, as implemented with the linregress function in the SciPy Stats module. Variance defaults (bivariate normal) for the function were not modified.

Overall Turnover by Watershed Crossings

To better understand changes in community as a function of topographic complexity, we delineated small watersheds in the area of the study area. The original digital elevation model used was the ASTER Global Digital Elevation Map version 2 [111]. Every subplot was assigned to a microwatershed and a distance matrix of watershed crossings. Watersheds were delineated using the pysheds package in Python [112]. The distance matrix of microwatershed crossings was calculated using Dijkstra's algorithm for shortest paths [113] in the NetworkX package version 2.4 [114]. Methods for specific watershed delineations are further explained in the statistical Jupyter Notebook (https://nbviewer.jupyter.org/github/danchurch/losCedrosTrees/blob/master/anaData/MariscalDataExploration.ipynb#watersheds, accessed on 31 April 2022).

### 2.3.3. Ordination by Historical Land Use/Habitat

Structuring in adult tree communities as a function of historical land use/habitat was visualized using the Bray–Curtis dissimilarity and nonmetric multidimensional scaling (NMDS) (Legendre and Legendre, 2012) with the metaMDS function in the Vegan package in R.

### 2.3.4. GIS Data and Additional Environmental Data

Except where otherwise noted, geospatial data were explored and visualized using tools from GeoPandas (version 0.8.1, https://geopandas.org, accessed on 31 April

2022), Rasterio (version 1.1.8, https://rasterio.readthedocs.io, accessed on 31 April 2022), and QGIS (QGIS v3.4.11-Madeira, OSGEO v3.7.1). Several environmental variables were generated using ASTER Global Digital Elevation Map version 2, including slope, aspect, elevation, eastern and northern exposures, and distance to nearest stream of all subplots. Stream data were digitized using Map CT-NII-C3-d of Bosque Protector Los Cedros from the Ecuadorian Instituto Geográfica Militar. Rasters of slope and aspect were generated with the Raster Terrain Analysis tool suite in QGIS.

### 2.3.5. Hierarchical Clustering of Sites

To understand the current types of forests present at Los Cedros, adult tree community data from all sites were partitioned into clusters using Ward's minimum variance clustering [106] using the Bray–Curtis dissimilarity, as implemented in the hclust command in the Vegan package in R. The number of clusters was decided by eye from the resulting dendrogram while taking into consideration the branch lengths and meaningful ecological groups.

### 2.3.6. Prediction of Current Ecological State by Land Use History and Elevation

The current ecological state, as defined based on the cluster analysis, was modeled as a function of historical land-use/habitat data and elevation. Each of the two predictors (height, land use/habitat) was considered individually, and then a combined model was created using both elevation and historical land use/habitat. All models were Bayesian and were written in Python using the PyMC3 package.

For land use/habitat, a multinomial logistic regression model was constructed with a softmax link function and the following priors:

$$\alpha \sim N(\mu = 0, \sigma = 5) \tag{12}$$

$$\beta \sim N(\mu = 0, \sigma = 5) \tag{13}$$

$$\theta = \text{softmax}(\alpha + \beta x) \tag{14}$$

$$y \sim \text{categorical}(\theta \,|\, y_{\text{observed}}) \tag{15}$$

where $y$ is a vector of predicted probabilities for forest type (cluster) of a site, distributed as a categorical random variable of $\alpha$, $\beta$, and $x$, and conditioned on our observed current forest type (i.e., cluster number); $x$ is the observed historical land use/habitat in a dummy variable matrix format; $\alpha$ is the y-intercept for each current forest type, and $\beta$ is the vector of slope coefficients for each of five types of historical land use/habitat for each current forest type. $\alpha$ and $\beta$ were assigned weak, normally distributed priors centered on zero.

Elevation: Hierarchical clustering of tree communities and NMS ordinations both indicated two distinct "natural" forest types occurring within the extent of the survey, which were designated as forest types III and IV (see the Results section). A logistic regression model was constructed to model differences among forest types III and IV from the hierarchical cluster analysis. To further understand differences between these two forest types, exploratory PERMANOVA models were used to test grouping by all available environmental predictors (see the Jupyter Notebook). Following this, the forest type was modeled as a function of elevation using a Bayesian logistic regression model. Priors and posterior were modeled as follows:

$$\alpha \sim N(\mu = 0, \sigma = 10) \tag{16}$$

$$\beta \sim N(\mu = 0, \sigma = 10) \tag{17}$$

$$\theta = \text{logistic}(\alpha + \beta x) \tag{18}$$

$$y \sim \text{Bernoulli}(\theta \,|\, y_{\text{observed}}) \tag{19}$$

where $\alpha$ and $\beta$ are the intercept and slope controlling the boundary decision between type III and type IV forests in terms of elevation; $\theta$ is the prior probability that a forest site will be a type III forest given its elevation; and y is a Bernoulli distribution with $\theta$ as the mean conditioned by the observed forest type.

For the combined elevation and historical land-use/habitat model, the elevation and historical land-use/habitat predictors were combined into one linear model:

$$\alpha \sim \mathrm{N}(\mu = 0, \sigma = 5) \tag{20}$$

$$\beta \sim \mathrm{N}(\mu = 0, \sigma = 5) \tag{21}$$

$$\Theta = \mathrm{softmax}(\alpha + \beta x) \tag{22}$$

$$y \sim \mathrm{categorical}(\theta \,|\, y_{\mathrm{observed}}) \tag{23}$$

where $x$ is the observed historical land use/habitat in a dummy variable matrix format, with an additional column vector giving the elevation of each site; $\alpha$ is the $y$-intercept for each row of $x$; and $\beta$ is the vector of slope coefficients for each column of $x$ for each current forest type (= row of $x$). $\alpha$ and $\beta$ were assigned weak, normally distributed priors centered on zero; y is a vector of predicted probabilities for forest type (cluster) of a site distributed as a categorical random variable of $\alpha$, $\beta$, and $x$, and conditioned on our observed current forest type (i.e., cluster number).

### 2.3.7. Indicator Species Analysis

Indicator species analyses were conducted to ascertain representative species for both historical land-use/habitat types and current forest type (cluster analysis results) using the indicspecies package in R [115] with the multipatt function and Pearson's phi coefficient correlation (function "r.g").

### 2.3.8. Spatial Analyses

To determine important spatial scales on which the tree community was changing, and to give shape to possible spatial structuring of the tree community, Moran's eigenvector maps (MEMs) were constructed [116] by following examples as found in documentation for the ADEspatial package (https://cran.r-project.org/web/packages/adespatial/vignettes/tutorial.html, accessed on 1 August 2020). A spatially weighted neighborhood matrix of sample sites was constructed using a Gabriel connectivity matrix with an inverse distance weighting. Abundances of our adult tree community matrix were Hellinger-transformed [106], and new PCA axes of the transformed were determined with the dudi.pca command in the ade4 package. Important MEMs were then selected based on their statistical significance as explanatory terms in a linear model, with the four most important tree community PCA axes as the response variables. Important MEMS were determined using a forward model-selection process, as implemented with the mem.select command in the adespatial package. The default cutoff of alpha = 0.05 was used to decide the statistical significance of MEMs. All available environmental variables were checked for correlations with MEMs using standard linear regression, as implemented in the linregress function in the Python SciPy Stats package. Correction for multiple testing was done with a Benjamini–Hochberg correction, as implemented in the multipletests command in the Python Statsmodels package. For exploratory purposes, the false discovery rate was set at FDR = 0.1.

### 2.3.9. Juvenile Communities

Juvenile tree community data were available for a subset of historical land-use/habitat types: "Bosque Cerrado (Closed Forest)" (BC), "Regeneración Caña de Azúcar (Regeneration from Sugar Cane)" (RCA), and "Regeneración de Fincas Agricultura y Ganadería (Regeneration from Agriculture and Pasture)" (RG). Juvenile data were not taken for "Claro del Bosque (Forest Clearings)" (CLB) or "Bosque Secundario (Secondary Forest)" (BS). Due

to the difficulties associated with identifying poorly understood herbaceous taxa and/or juvenile trees, these data required extensive additional cleaning and verification following the botanical identifications. For the purposes of this study, juvenile trees were defined as undersized plants that would, in time and with proper growth conditions, reach the stature of the mature trees as defined above; namely, woody plants with a diameter-at $-1.5$ m of 10 cm or greater. This was as compared to herbaceous plants or small woody plants (such as lianas or subshrubs), which would never or rarely reach a 10 cm girth at 1.5 m height.

Samples that did not receive sufficient identification to confidently conclude that the sample was indeed a juvenile tree were discarded from the analysis. This selection was done using automated and manual queries of growth form data requested from the TRY database [117], manual checks of the Encyclopedia of Life [118], and the Smithsonian Tropical Research collections site (https://stricollections.org/portal/index.php, accessed on 1 October 2020). The Gentry manual [119] was also frequently consulted. Exact species information was often not available, and the designation of juvenile tree status was made using genus growth form data where possible. Juvenile tree communities were compared to adult trees by subsetting adult tree community data to just those sites with juvenile data and combining community matrices from both. These were then ordinated using the Bray–Curtis dissimilarity and nonmetric multidimensional scaling (NMS) via the metaMDS function in the Vegan package in R.

### 2.3.10. Deforestation in the Region

To understand the extractive pressure in the region of Los Cedros, as well as the efficacy of Los Cedros as a conservation project, we examined land-cover changes from 1990 to 2018. Data on changing land cover were accessed from the Ecuadorian Ministerio del Ambiente's ("MAE") interactive map and GIS catalogue, using the "Cobertura vegetal" layers from 1990 and 2018 (available at https://gis-sigde.maps.arcgis.com/apps/webappviewer/idex.html?id=8b53f9388c034b5e8e3147f03583d7ec&fbclid=IwAR2XobS46Szpz4A7IGroPuLCZh5GSJC, accessed on 10 April 2021). Vector data from the MAE were rasterized to a pixel resolution of 30 m$^2$ using the gdal_rasterize program in the GDAL tool suite of the OSGeo project [120] and visualized in Matplotlib using the rasterio package. Background deforestation rates were constrained to the Cotacachi canton of the Imbabura province, in which Los Cedros is located. Deforestation rates in the similarly-sized, nearby Bosque Protector el Chontal were quantified for comparison with Los Cedros. BP Chontal is managed by the Cattlemen's Association of the nearby town of Brillasol (pers comm. Jose DeCoux). Administrative boundaries of BP Los Cedros and BP Chontal were supplied using the official boundary layer "Bosque y Vegetación protectora", also available at MAE's interactive map website.

## 3. Results

Pertinent posterior distributions are described here. Full results of the posterior distributions of all models are available in the Jupyter Notebook.

### 3.1. Species Accumulation Curves and Richness Estimators

A total of 343 adult tree species were directly observed in plots. Estimates of total adult tree species in the study area range from 404 to 566 species (Figure 1A, Table 1). In the permanent 0.5 ha plot, 43 species of tree were observed, distributed among 36 genera and 25 plant families, with a range of predicted total species from 51 to 72 species in the half-hectare (Figure 1B, Table 1). On average, 1 ha in the southern portion of Los Cedros was predicted to host 169 species of adult tree. Chao estimates, first- and second-order jackknife, and bootstrap estimates for the total study area and for the permanent 0.5 ha plot are given in Table 1.

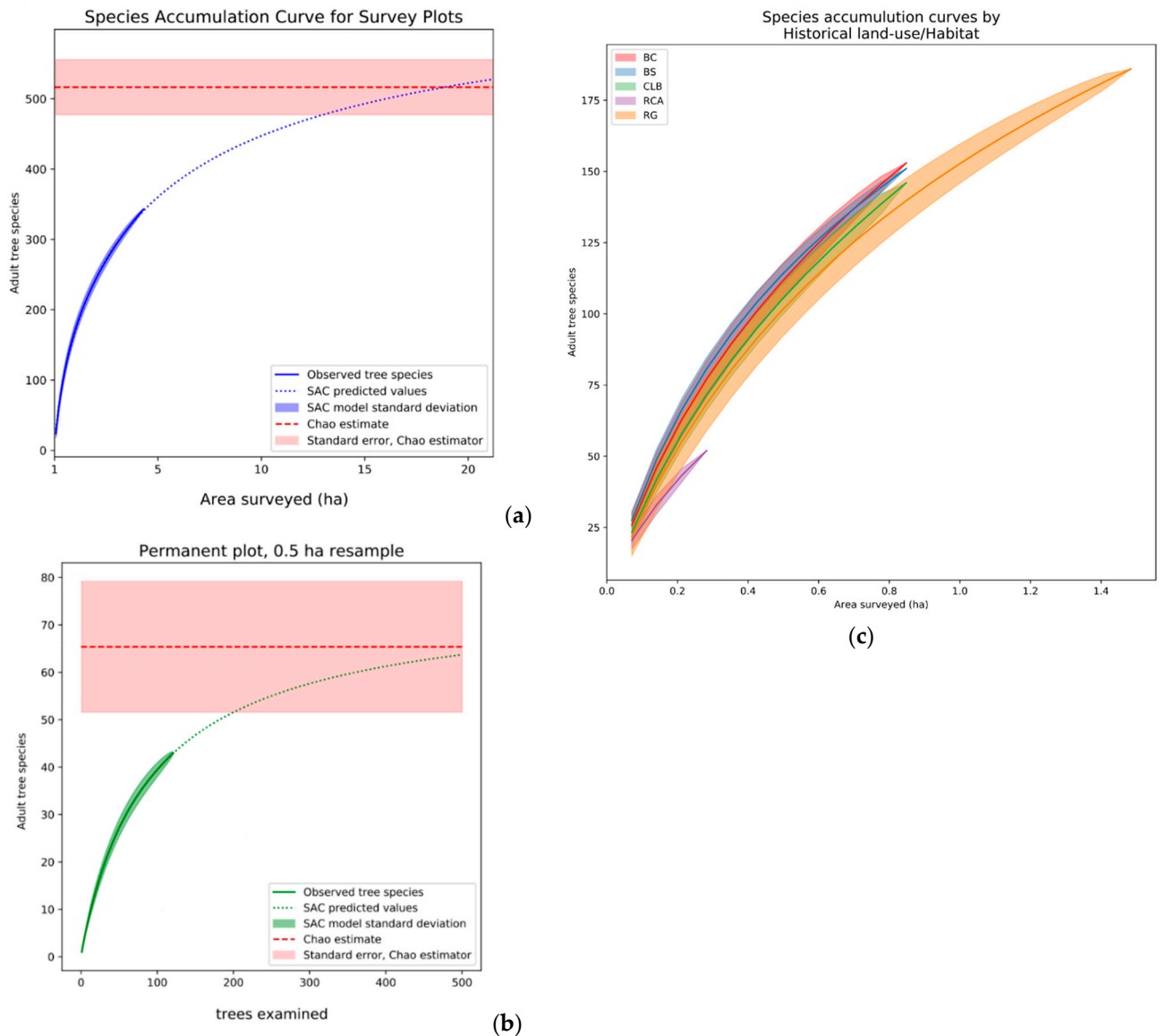

**Figure 1.** Species accumulation curves (SACs): (**a**) SAC for entirety of study, with all land uses combined. Dotted line is the estimated number of species using a Chao estimator, with one standard error. (**b**) SAC for single 0.5 hectare permanent plot. Dotted line is the estimated number of species using a Chao estimator, with one standard error. (**c**) SAC for entirety of study, itemized by land-use/habitat type. See Section 2.2.1 for explanation of land-use/habitat abbreviations.

**Table 1.** Estimates of adult tree species diversity in survey area and permanent plot, Using Chao, First- and Second-order jackknife, and bootstrap methods. Uncertainties are given as standard error (".Se").

|  | Sample Size | Species Observed | Chao | Chao.Se | Jack1 | Jack1.Se | Jack2 | Boot | Boot.Se |
|---|---|---|---|---|---|---|---|---|---|
| Survey Plots | 61 | 343 | 516.4 | 39.2 | 483.7 | 21.6 | 566.8 | 404.4 | 10.5 |
| Permanent Plot (Trees) | 121 | 43 | 65.4 | 13.8 | 61.8 | 4.3 | 72.7 | 51.4 | 2.2 |

When examined by land-use/habitat type, more adult tree species were observed in reforested pasture sites, possibly due to deeper sampling, but natural-disturbance sites (BS, BC, and CLB) were comparable when rarified to a common sample size (Figure 1C, Supplementary Table S2). Sites with a history of intensive agricultural use were the least diverse, though the depth of sampling was lowest in this group. Species estimators by land-use history are given in Supplementary Table S2.

### 3.2. Tree Community Turnover (Distance Decay)

3.2.1. Turnover by Physical Distance

1. Asymptotic model

The asymptote model was visualized with the 50% and 95% highest posterior densities ("HPD") for the predicted Bray–Curtis values, as shown in Figure 2A. The posterior distribution of our asymptotic model explained a large amount of the variance in the adult tree community data ($R^2 = 0.53 \pm 0.11$). The distribution of posterior predictive values was nearly symmetric around the observed Bray–Curtis values (Bayesian *p*-value = 0.49, Supplementary Figure S2). The posterior distribution of $K_m$ was centered on 153 m ($\pm$ 6.13 m), a shift of $-47$ m from the prior estimate of 200 m, with no overlap between the posterior and prior (Supplementary Figure S3).

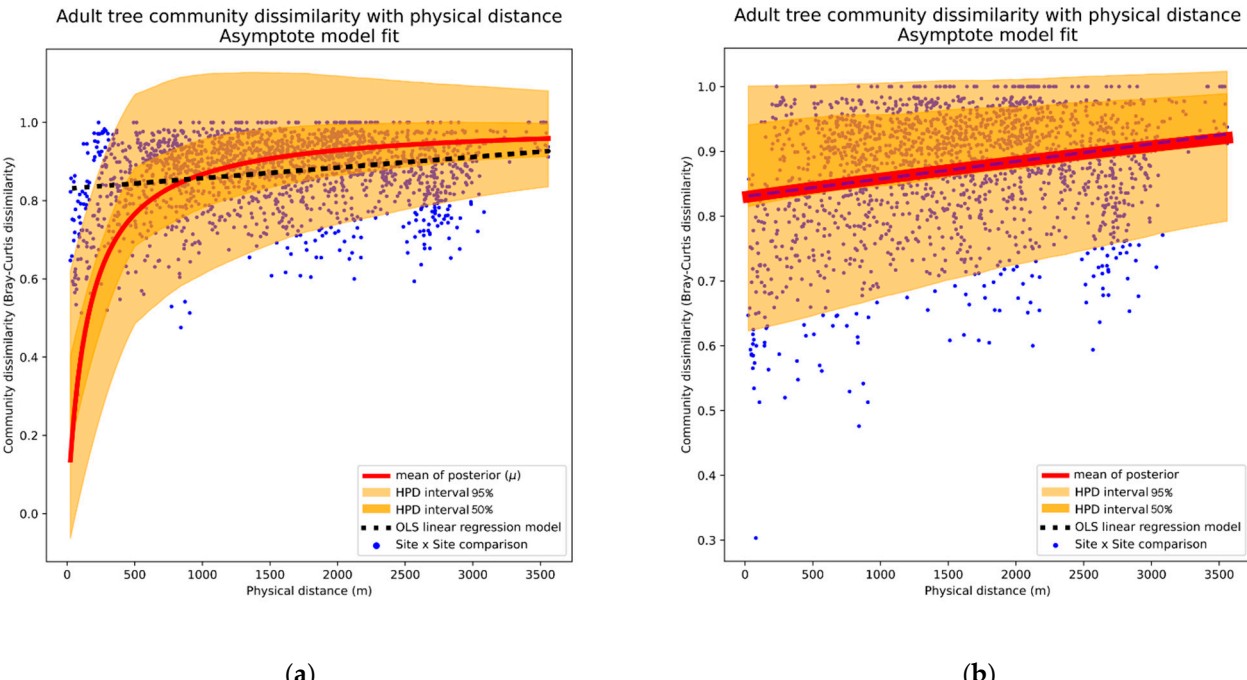

(**a**)                                                                 (**b**)

**Figure 2.** Community turnover and community dissimilarity models. Blue points represent Bray–Curtis dissimilarities between tree communities among all 61 sites, with increasing physical distance between sites being compared. Two approaches are considered, an asymptotic model and a simple linear model. (**a**) Bayesian, asymptotic (Michaelis–Menten) model fit. In this model, the function of the mean community dissimilarity value of zero was enforced when comparing sites that were nearly zero meters apart, and asymptotically approached complete dissimilarity (Bray–Curtis dissimilarity = 1). For comparison, the classical ordinary-least-squares fit is also given (black dashed line). (**b**) Bayesian linear model fit.

2. Linear model

The Bayesian linear model was visualized with the 50% and 95% highest posterior densities ("HPD") for the predicted Bray–Curtis values, as shown in Figure 2B. As constructed, the posterior distribution of our linear model explained only a small amount of the variance in the adult tree community data ($R^2 = 0.06 \pm 0.01$). The posterior predictive values tended to be lower than observed Bray–Curtis values (Bayesian *p*–value = 0.41, Supplementary Figure S2). The frequentist, least-squares linear regression model highly resembled the Bayesian model, also reporting very little variance explained ($R^2 = 0.06 \pm 0.01$, *p* = < 0.01, Figure 2B).

### 3.2.2. Overall Turnover by Watershed Crossings

When plotted as a function of watershed crossings (Figure 3), the widest range of Bray–Curtis values occurred among sites within the same watershed (distance class 0), as did the lowest mean Bray–Curtis score (mean Bray–Curtis dissimilarity = 0.82, ± 0.11). Following this, other mean Bray–Curtis values of comparisons of other distance class did not vary heavily, and were mostly statistically indistinguishable (Kruskal–Wallis H-test (H (5, 1825) = 73.27, $p < 0.01$; and Tukey HSD ($p$–adj = 0.001); Tables 2 and 3, Figure 3). After the first watershed crossing, further comparisons were all approximately equally dissimilar, with a mean of Bray–Curtis = 0.88 (±0.086).

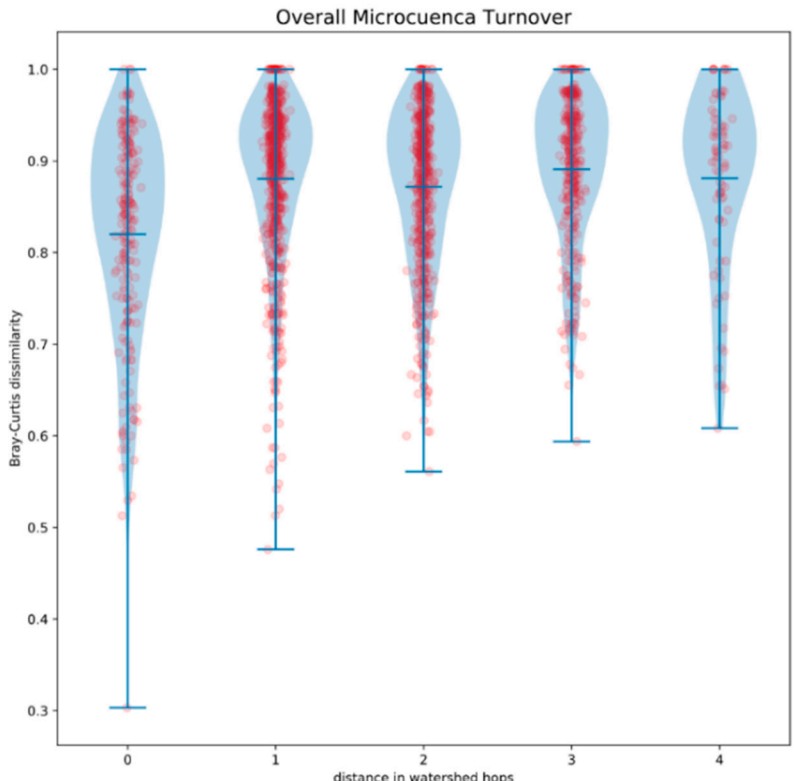

**Figure 3.** Binning of community turnover into watershed crossings ("hops"). Red points represent Bray–Curtis dissimilarities between tree communities among all 61 sites with increasing watershed crossings between sites.

**Table 2.** Mean dissimilarity (Bray-Curtis dissimilarity) of survey sites by number of watersheds crossed.

| Watersheds Crossed | Sample Size | Mean BC | Std. dev BC |
|---|---|---|---|
| 0 | 215 | 0.820 | 0.114 |
| 1 | 630 | 0.880 | 0.089 |
| 2 | 559 | 0.872 | 0.085 |
| 3 | 346 | 0.891 | 0.079 |
| 4 | 80 | 0.881 | 0.096 |

**Table 3.** Tukey's honest significant difference of comparisons among watershed distance classes.

| Group1 | Group2 | Meandiff | *p*-Adj | Lower | Upper | Reject |
|--------|--------|----------|---------|-------|-------|--------|
| 0 | 1 | 0.0601 | 0.001 | 0.0408 | 0.0795 | TRUE |
| 0 | 2 | 0.0516 | 0.001 | 0.0319 | 0.0713 | TRUE |
| 0 | 3 | 0.0708 | 0.001 | 0.0495 | 0.0921 | TRUE |
| 0 | 4 | 0.0609 | 0.001 | 0.0287 | 0.093 | TRUE |
| 1 | 2 | −0.0085 | 0.4759 | −0.0228 | 0.0057 | FALSE |
| 1 | 3 | 0.0106 | 0.393 | −0.0058 | 0.0271 | FALSE |
| 1 | 4 | 0.0007 | 0.9 | −0.0284 | 0.0299 | FALSE |
| 2 | 3 | 0.0192 | 0.0158 | 0.0024 | 0.036 | TRUE |
| 2 | 4 | 0.0093 | 0.9 | −0.0201 | 0.0386 | FALSE |
| 3 | 4 | −0.0099 | 0.9 | −0.0404 | 0.0205 | FALSE |

*3.3. Hierarchical Clustering of Sites and Prediction of Current Ecological State by Land-Use History and Elevation*

Tree communities were first sorted into two large groups that aligned well with previous land use: the first grouping contained sites that have historically experienced anthropogenic disturbances of any type, and the second grouping contained sites with no recorded history of anthropogenic disturbance, but all types of natural gap-forming disturbances (Figure 4). Each of these groups then were sorted into two further groups (for a total of four clusters). Cluster I contained all sites considered to be highly disturbed by agricultural use (RCA sites) and some sites with intermediate agricultural use as pasture (RG sites) (Figure 4). Cluster II consisted entirely of RG sites, and also contained the majority of RG sites. Sites with no history of anthropogenic disturbances fell into two clusters: type III and type IV (Figure 4). The hierarchical clustering results were congruent with the NMS ordinations (Figure 5, Supplementary Figure S4). A summary map giving both the current ecological state and the historical land use/habitat is given in Figure 6.

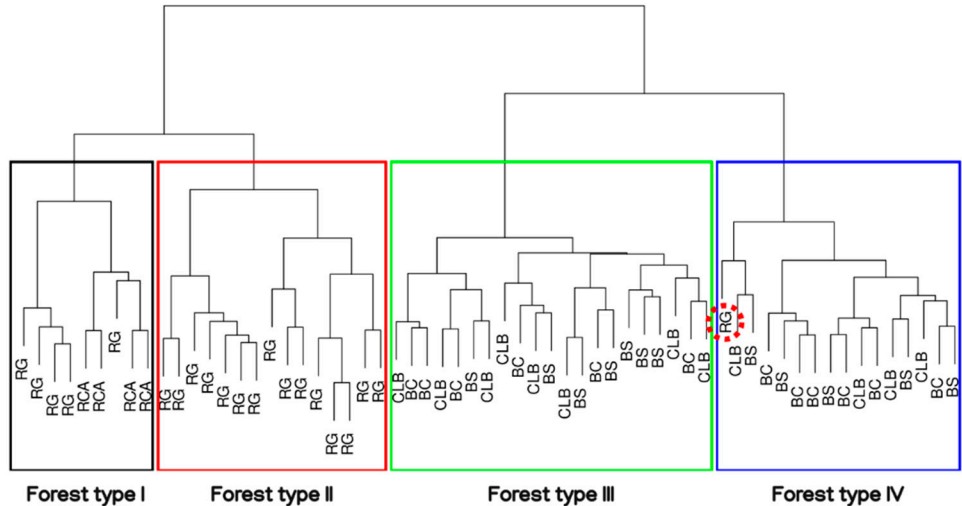

**Figure 4.** Hierarchical clustering of all sites using species community composition. Tips of the tree represent sites. These four obvious groupings were used to categorize the current ecological state of the forest at each site, or "Forest type". Site circled with a red-dotted line represents site 10.1, which was historically pastured but regenerated to resemble an "old" forest type IV. The history of land use at each site is noted.

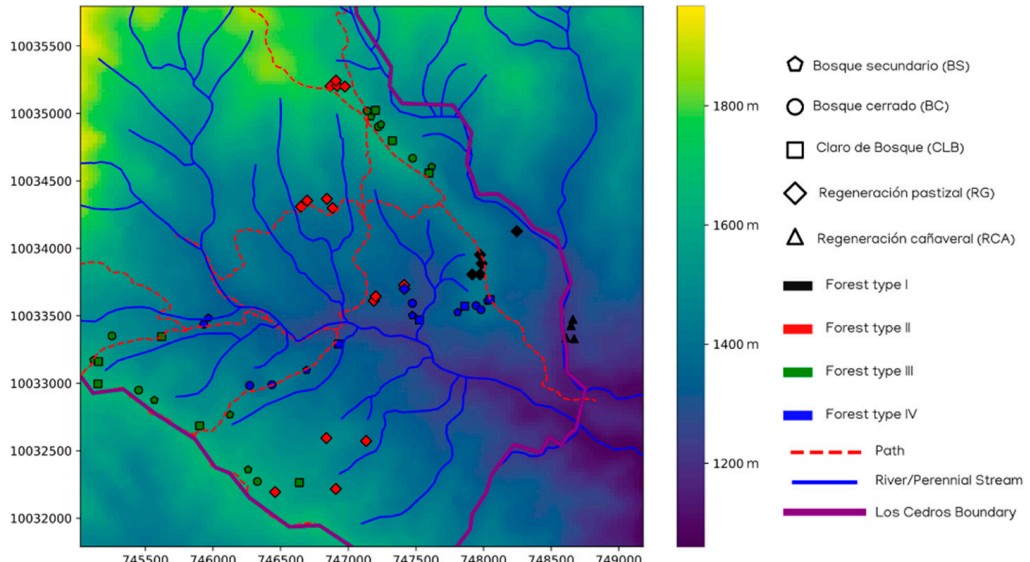

**Figure 5.** Nonmetric multidimensional scaling (NMS) ordination of adult tree communities using Bray–Curtis dissimilarity. Kruskal's stress = 0.29. Both land-use history (hulls) and current ecological state ("Forest Type", point fill color) are noted. Site circled with a red-dotted line represents site 10.1, which was historically pastured but regenerated to resemble an "old" forest type IV.

**Figure 6.** Map of sites with land-use history and current ecological state ("forest type"). *X* and *Y* coordinates are Universal Transverse Mercator (zone 17S) coordinates in meters.

A linear model with these ecological states (forest types I–IV) as the response variable using the single variable of previous land-use/habitat as predictors successfully predicted the sites that had been modified for pasture or agriculture, and it also well predicted nonanthropogenically disturbed sites as a type III/IV ecological state ($R^2 = 0.65 \pm 0.04$, Supplementary Figure S5). Posterior predictive checks were correct for 65% of sites, mostly for sites with type I and type II forests. However, this previous land-use/habitat-only model could not distinguish among "natural forest" (III and IV) types.

Among sites with no history of anthropogenic disturbance ("natural forests", types III and IV), elevation strongly predicted the current ecological category of forest: sites with no history of anthropogenic disturbance at elevations of 1503 m (95% credible interval = 1469 m to 1537 m) or higher were categorized almost completely as type III forests, and below this elevation, sites were categorized as type IV forests (Bayesian logistic regression, $R^2 = 0.87$; Figures 6 and 7, Supplementary Figure S6).

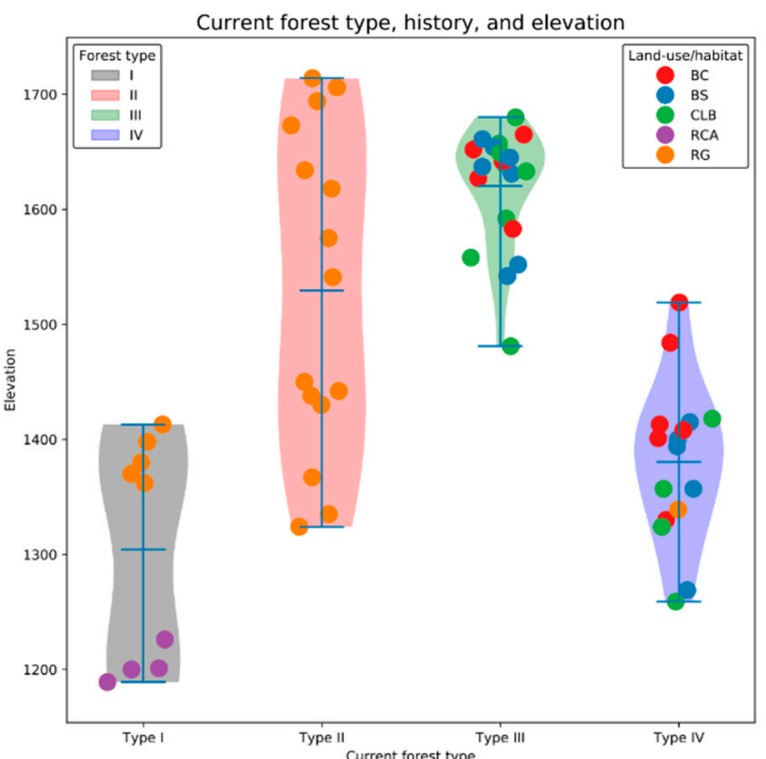

**Figure 7.** Current ecological state of sites ("forest type") and their elevations. Historical land uses of sites are indicated by color of point fill.

A combined model of previous land use/habitat and elevation explained most of the variance in the total and predicted the current ecological state very well (Bayesian linear model, $R^2 = 0.88 \pm 0.16$; Supplementary Figure S7), and performed much better in the posterior predictions, with 87% of sites correctly predicted as their forest type.

### 3.4. Indicator Species Analysis

Indicator species were detected for all individual land-use-history/habitat types, all current ecological states (forest types), and for some combinations of site types. The indicator species analysis results are given in Table 4.

**Table 4.** Indicator species for historical land use/habitat, and indicator species for current ecological state (forest type). "Rpb.G" represents the point-biserial-serial-correlation-coefficient, a measure of the strength of the co-occurrence pattern observed between a tree species and a habitat-type or ecological state. Statistical significance codes: "***" $p < 0.001$; "**" $p < 0.01$; "*" $p < 0.05$.

| Indicator Species by Land-Use-History/Habitat-Type | | | |
|---|---|---|---|
| **SGroup BC** | | | |
| Indicator Species | Rpb.G | *p*-Value | Sig Code |
| *Pseudolmedia rigida* (Moraceae) | 0.423 | 0.0317 | * |
| *Ficus subandina* (Moraceae) | 0.406 | 0.049 | * |
| **Group BS** | | | |
| | Rpb.G | *p*-Value | Sig Code |
| *Psychotria paeonia* (Rubiaceae) | 0.478 | 0.0148 | * |
| *Persea pseudofasciculata* (Lauraceae) | 0.432 | 0.0351 | * |
| *Myrcia* aff. *aliena* (Myrtaceae) | 0.425 | 0.0377 | * |
| **Group CLB** | | | |
| | Rpb.G | *p*-Value | Sig Code |
| *Endlicheria* cf. *chalisea* (Lauraceae) | 0.431 | 0.0268 | * |
| **Group RCA** | | | |
| | Rpb.G | *p*-Value | Sig Code |
| *Cordia colombiana* (Boraginaceae) | 0.894 | 0.0001 | *** |
| *Saurauia* sp. 1 (Actinidaceae) | 0.694 | 0.0003 | *** |
| *Miconia* aff. *brevitheca* (Melastomataceae) | 0.683 | 0.0004 | *** |
| *Ficus caldasiana* (Moraceae) | 0.667 | 0.0033 | ** |
| *Turpinia occidentalis* (Staphyleaceae) | 0.663 | 0.0006 | *** |
| *Clarisia biflora* (Moraceae) | 0.645 | 0.0022 | ** |
| *Nectandra membranacea* (Lauraceae) | 0.61 | 0.0027 | ** |
| *Aegiphila alba* (Verbenaceae) | 0.594 | 0.002 | ** |
| *Ficus andicola* (Moraceae) | 0.539 | 0.0191 | * |
| *Caryodaphnopsis theobromifolia* (Lauraceae) | 0.535 | 0.0099 | ** |
| *Nectandra* aff. *membranacea* (Lauraceae) | 0.513 | 0.0106 | * |
| **Group RG** | | | |
| | Rpb.G | *p*-Value | Sig Code |
| *Cecropia andina* (Cecropiaceae) | 0.662 | 0.0018 | ** |
| *Cecropia ficifolia* (Cecropiaceae) | 0.58 | 0.0035 | ** |
| *Cecropia* sp. 2 (Cecropiaceae) | 0.427 | 0.031 | * |
| *Meriania tomentosa* (Melastomataceae) | 0.415 | 0.0426 | * |
| **Group BC + BS** | | | |
| | Rpb.G | *p*-Value | Sig Code |
| *Aniba* aff. *hostmanniana* (Lauraceae) | 0.411 | 0.0471 | * |
| **Group RCA + RG** | | | |
| | Rpb.G | *p*-Value | Sig Code |
| *Solanum lepidotum* (Solanaceae) | 0.529 | 0.006 | ** |

**Table 4.** *Cont.*

| Indicator Species by Land-Use-History/Habitat-Type | | | |
|---|---|---|---|
| *Cestrum megalophyllum* (Solanaceae) | 0.423 | 0.0268 | * |
| Group BC + BS + CLB | | | |
| | Rpb.G | *p*-Value | Sig Code |
| *Alsophila erinacea* (Cyatheaceae) | 0.42 | 0.0423 | * |
| Indicator Species by Current Ecological State (Forest Type) | | | |
| Group I | | | |
| | Rpb.G | *p*-Value | Sig Code |
| *Cordia colombiana* (Boraginaceae) | 0.661 | 0.0001 | *** |
| *Meriania tomentosa* (Melastomataceae) | 0.641 | 0.0001 | *** |
| *Saurauia* sp. 1 (Actinidaceae) | 0.545 | 0.0003 | *** |
| *Cyathea halonata* (Cyatheaceae) | 0.458 | 0.0033 | ** |
| *Senna dariensis* (Fab. Caesalpiniaceae) | 0.449 | 0.0046 | ** |
| *Ficus caldasiana* (Moraceae) | 0.42 | 0.0206 | * |
| *Leandra subseriata* (Melastomataceae) | 0.37 | 0.015 | * |
| *Miconia* aff. *brevitheca* (Melastomataceae) | 0.368 | 0.027 | * |
| *Piper fuliginosum* (Piperaceae) | 0.356 | 0.05 | * |
| *Turpinia occidentalis* (Staphyleaceae) | 0.346 | 0.0426 | * |
| Group II | | | |
| | Rpb.G | *p*-Value | Sig Code |
| *Cecropia andina* (Cecropiaceae) | 0.811 | 0.0001 | *** |
| *Cecropia* sp. 2 (Cecropiaceae) | 0.5 | 0.0009 | *** |
| Melastomataceae sp. 1 | 0.468 | 0.0051 | ** |
| *Dussia lehmannii* (Fab. Faboideae) | 0.334 | 0.0444 | * |
| Group III | | | |
| | Rpb.G | *p*-Value | Sig Code |
| *Otoba gordoniifolia* (Myristicaceae) | 0.524 | 0.0007 | *** |
| *Alsophila erinacea* (Cyatheaceae) | 0.515 | 0.0015 | ** |
| *Persea* aff. *pseudofasciculata* (Lauraceae) | 0.494 | 0.0022 | ** |
| *Vismia lauriformis* (Clusiaceae) | 0.44 | 0.0065 | ** |
| *Conostegia* aff. *centronioides* (Melastomataceae) | 0.418 | 0.0083 | ** |
| *Wettinia* aff. *oxycarpa* (Arecaceae) | 0.403 | 0.0115 | * |
| *Persea pseudofasciculata* (Lauraceae) | 0.386 | 0.0217 | * |
| *Psychotria paeonia* (Rubiaceae) | 0.373 | 0.0264 | * |
| *Ficus dulciaria* (Rubiaceae) | 0.354 | 0.03 | * |
| Group IV | | | |
| | Rpb.G | *p*-Value | Sig Code |
| *Dacryodes cupularis* (Burseraceae) | 0.754 | 0.0001 | *** |
| *Protium ecuadorense* (Burseraceae) | 0.638 | 0.0001 | *** |
| *Garcinia macrophylla* (Clusiaceae) | 0.613 | 0.0001 | *** |

**Table 4.** *Cont.*

| Indicator Species by Land-Use-History/Habitat-Type | | | |
| --- | --- | --- | --- |
| *.Beilschmiedia* aff. *costaricensis* (Lauraceae) | 0.448 | 0.0018 | ** |
| *Conostegia superba* (Melastomataceae) | 0.444 | 0.0024 | ** |
| *Ocotea stenoneura* (Lauraceae) | 0.369 | 0.0454 | * |
| *Gustavia dodsonii* (Lecythidaceae) | 0.362 | 0.029 | * |
| *Styrax weberbaueri* (Styracaceae) | 0.361 | 0.0408 | * |
| *Pseudolmedia rigida* (Moraceae) | 0.355 | 0.0263 | * |
| Group I + II | | | |
| | Rpb.G | *p*-Value | Sig Code |
| *Cecropia ficifolia* (Cecropiaceae) | 0.52 | 0.0005 | *** |
| *Solanum lepidotum* (Solanaceae) | 0.447 | 0.0055 | ** |
| *Cecropia reticulata* (Cecropiaceae) | 0.371 | 0.0244 | * |
| *Aegiphila alba* (Verbenaceae) | 0.348 | 0.0465 | * |
| *Urera caracasana* (Urticaceae) | 0.341 | 0.037 | * |
| Group I + IV | | | |
| | Rpb.G | *p*-Value | Sig Code |
| *Caryodaphnopsis theobromifolia* (Lauraceae) | 0.416 | 0.0092 | ** |
| *Clarisia biflora* (Moraceae) | 0.403 | 0.0157 | * |
| Group II + III | | | |
| | Rpb.G | *p*-Value | Sig Code |
| *Guatteria megalophylla* (Annonaceae) | 0.411 | 0.0133 | * |
| *Ficus cuatrecasana* (Moraceae) | 0.364 | 0.0299 | * |
| Group III + IV | | | |
| | Rpb.G | *p*-Value | Sig Code |
| *Aniba* aff. *hostmanniana* (Lauraceae) | 0.401 | 0.0138 | * |
| *Capparis* sp. (Capparaceae) | 0.385 | 0.0267 | * |
| *Hieronyma asperifolia* (Euphorbiaceae) | 0.383 | 0.0226 | * |
| *Eschweilera integrifolia* (Lecythidaceae) | 0.379 | 0.0199 | * |
| *Ocotea insularis* (Lauraceae) | 0.366 | 0.0295 | * |
| *Faramea oblongifolia* (Rubiaceae) | 0.362 | 0.0291 | * |
| *Helicostylis tovarensis* (Moraceae) | 0.346 | 0.0444 | * |

*3.5. Spatial Analyses*

A total of 13 Moran's eigenvector maps variables were detected as correlated with changes in tree community (Figure 8; cumulated, adjusted $R^2 = 0.19$, all MEMs $p < 0.02$ as components in forward model selection). Most were substantially correlated with 1–3 available environmental variables or a particular land-use/habitat history (Table 5), with the exception of one highly localized MEM variable, MEM3. The most influential MEM variables in the model, MEM1 and MEM2, presented obvious geographic patterns. MEM1 (contributing $R^2 = 0.026$ to cumulative $R^2$, or 13.9% of explainable variance) indicated a general difference in tree communities between those of the northeastern and southwestern sides of the Los Cedros River (Supplementary Figure S8). The second most influential MEM variable, MEM2 (contributing $R^2 = 0.026$ to cumulative $R^2$, or 13.6% of explainable

variance), indicated a difference between the highest sites sampled in the study, which were situated along a ridge that runs ultimately to the highest points in the reserve, and lower elevations along this ridge system (Supplementary Figure S9). MEM8 also was correlated with elevation and with distance to nearest stream, likely indicating a difference among "high and dry" sites and lower, wetter sites (Supplementary Figure S10).

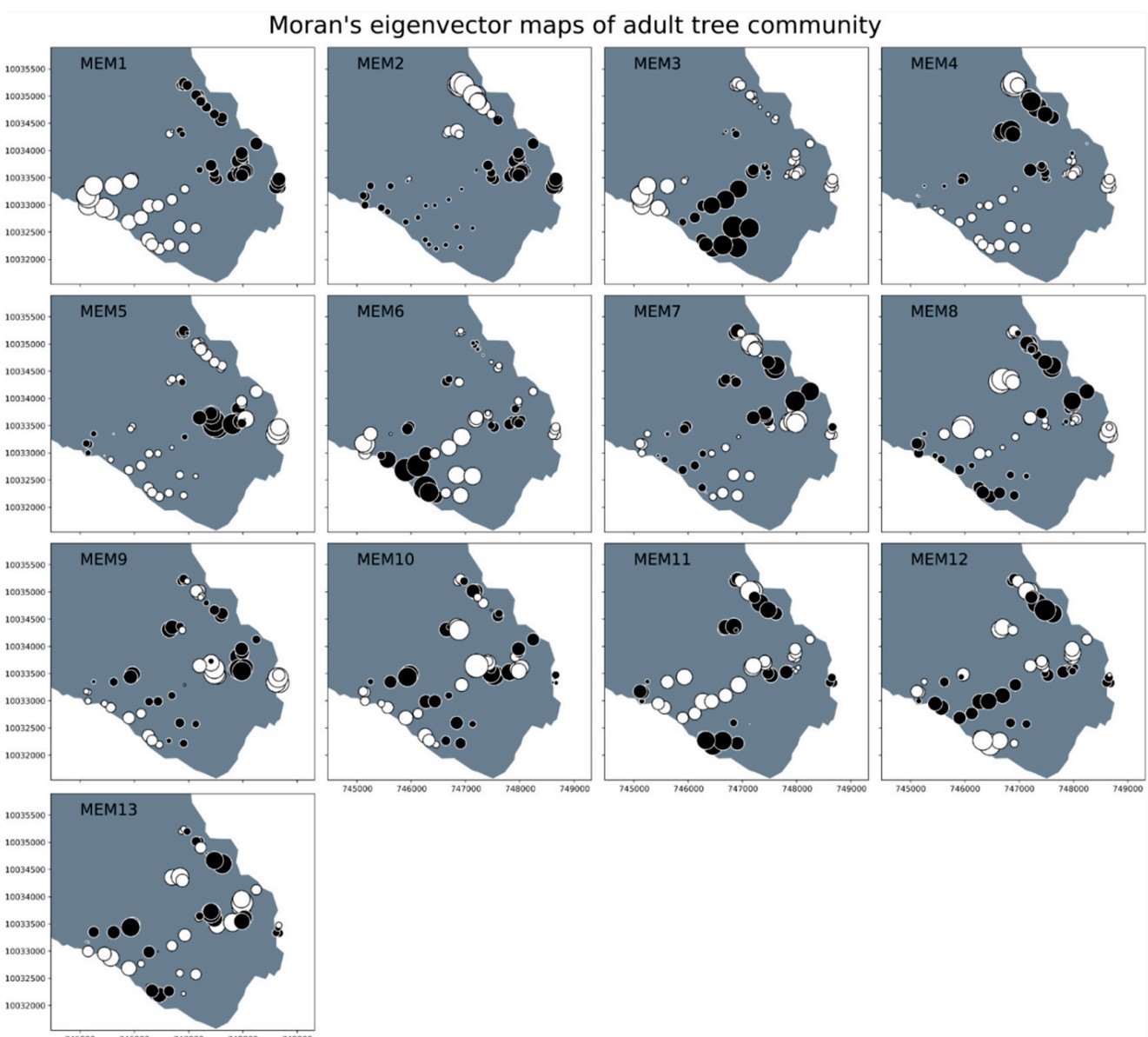

**Figure 8.** All statistically significant Moran's eigenvector maps (MEMs) detected from tree community data. Within each map, two sites of large size but differing colors were maximally different (i.e., a site with a large white circle had some large difference in species composition with a site with a large black circle). Correlations between these spatial patterns and available environmental data are given in Table 5.

**Table 5.** Moran's eigenvector maps and their statistically significant environmental correlations.

|  | Slope | Dem | Aspect | Exposure (Eastern) | Exposure (Northen) | Tostream | BC | BS | CLB | RCA | RG |
|---|---|---|---|---|---|---|---|---|---|---|---|
| MEM1 |  | −0.403 |  |  |  |  |  |  |  | 0.287 |  |
| MEM2 | −0.344 | −0.636 |  |  |  |  |  |  |  | 0.308 |  |
| MEM3 |  |  |  |  |  |  |  |  |  |  |  |
| MEM4 |  |  |  |  | 0.349 |  |  |  |  |  |  |
| MEM5 |  |  |  |  |  |  |  |  |  | −0.503 |  |
| MEM6 |  |  |  |  |  | 0.306 |  |  |  |  |  |
| MEM7 |  |  |  |  |  |  |  |  |  |  | 0.385 |
| MEM8 |  | 0.404 |  |  |  | 0.460 |  |  |  |  |  |
| MEM9 |  |  |  |  |  |  |  |  |  | −0.451 |  |
| MEM10 |  |  |  |  |  | −0.340 |  |  |  |  |  |
| MEM11 |  |  |  |  |  | 0.282 |  |  |  |  |  |
| MEM12 |  |  |  |  |  |  |  |  |  |  | −0.410 |
| MEM13 |  |  |  |  |  |  | 0.345 |  |  |  |  |

*3.6. Juvenile Communities*

Juvenile communities did not cluster within the groups presented by adult communities, neither in terms of historical land use/habitat (Figure 9A) nor current ecological state of adult trees (Figure 9B). Rather, all juvenile communities radiated into new dissimilarity space from their respective adult communities, possibly indicating recruitment of new tree species and/or new species combinations at these sites. Of the 148 species of juvenile tree species observed in the study, 110 species were observed as both juvenile and mature specimens, and an additional 38 species were observed only as juvenile specimens.

Juvenile tree communities changed along the same axes as were observed for their respective adult tree communities. When categorized using the historical land-use/habitat variable, RCA sites and related, similar RG sites were pulled in a negative direction along the NMS1 axis, as shown in Figure 9A, and nonanthropogenically disturbed sites (BC, BS, and CLB) were pulled in a positive direction along the NMS2 axis. When categorized according to the current ecological state of their adult tree communities, the same pattern held (Figure 9B): juvenile tree communities were different and outside of clusters formed by their respective adult communities, but changed along the same axes as their adult communities. Two exceptions to this were observed: the juvenile tree communities of two type I forest sites have come to more closely resemble type III (Figure 9B, sites 1.2 and 1.3). These sites had a land-use/habitat history of pasture conversion (RG), and their adult tree communities clustered into a current ecological state of type I forest, the most anthropogenically disturbed group, and often associated with RCA land-use history (Figure 5). However, their juvenile tree communities now resemble more "natural" forest types. Additionally, the site 1.1 juvenile tree community was so unique as to be an outlier to the rest of the sites studied, and had to be removed for informative examination of the remaining sites (Supplementary Figure S11).

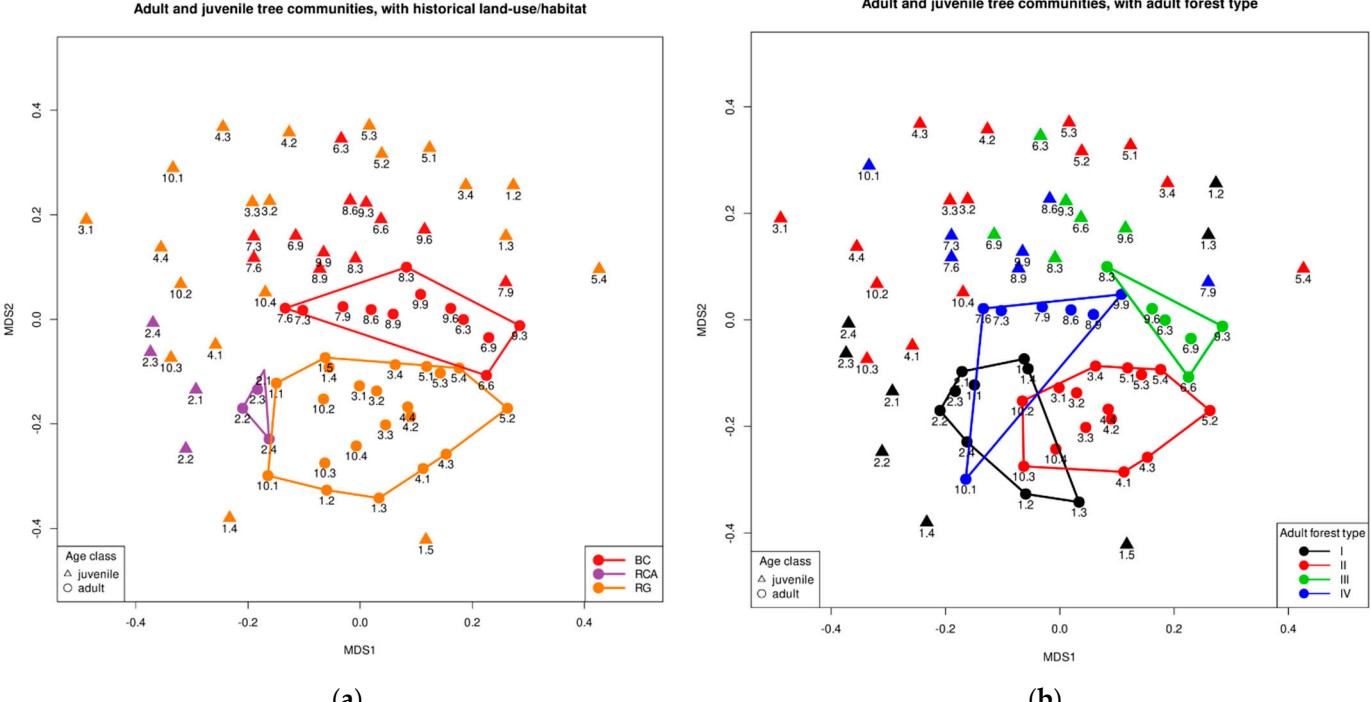

**Figure 9.** Nonmetric multidimensional scaling (NMS) ordination of combined juvenile and adult tree communities using Bray–Curtis dissimilarity. Only sites with both juvenile and adult data are shown. Each site therefore had two points, a juvenile (triangle) and adult (circle) tree community. (**a**) Ordination colored by historical land use. Hulls are drawn around adult communities. (**b**) Ordination colored by current ecological state ("forest type"). Hulls are drawn around adult communities.

### 3.7. Deforestation in the Region

Between 1990 and 2018, forest cover in the Cotacachi canton was reduced from an estimated 87,967 ha of native forest cover in 1990 to 71,739 ha in 2018, an 18% reduction in the total forest from 1990 levels, mostly due to conversion to agricultural land (Figure 10). In terms of total land cover, this equated to a shift from approximately 52% of total Cotacachi canton land cover being native forest to 42% of total land cover as native forest. Forest cover in Los Cedros increased from an estimated 5094 ha of native forest cover in 1990 to 5210 ha in 2018, a 2.3% addition to the total forest at 1990 levels, due mostly to the reforestation of former pasture (Figure 11). Forest cover in Bosque Protector Chontal decreased from an estimated 6920 ha of native forest cover in 1990 to 6565 ha in 2018, a 5% reduction in the total forest from 1990 levels, due mostly to conversion of forest to agricultural land uses (Figure 11).

Unlike earlier datasets, the 2018 MAE land cover dataset does not include pasture separately from other forms of agricultural land use, so it was difficult to quantify which types of agricultural land use were most commonly replacing forest in the region. However, based on visitation to the surrounding communities, it is the authors' assessment that much of the recent deforestation was undertaken to create pasture for cattle grazing.

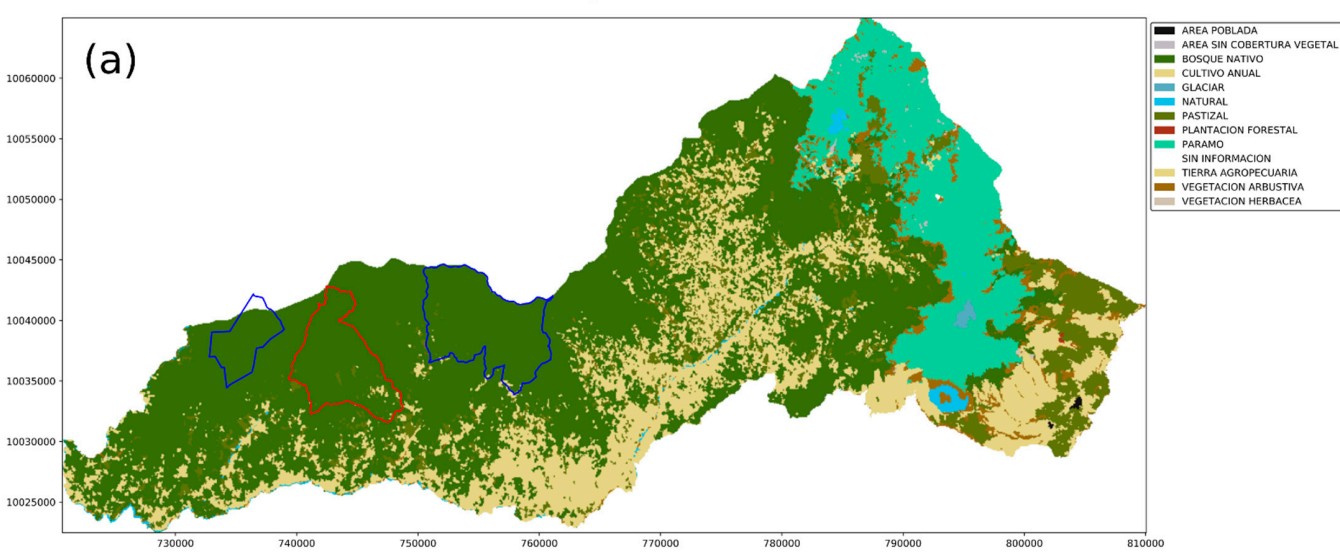

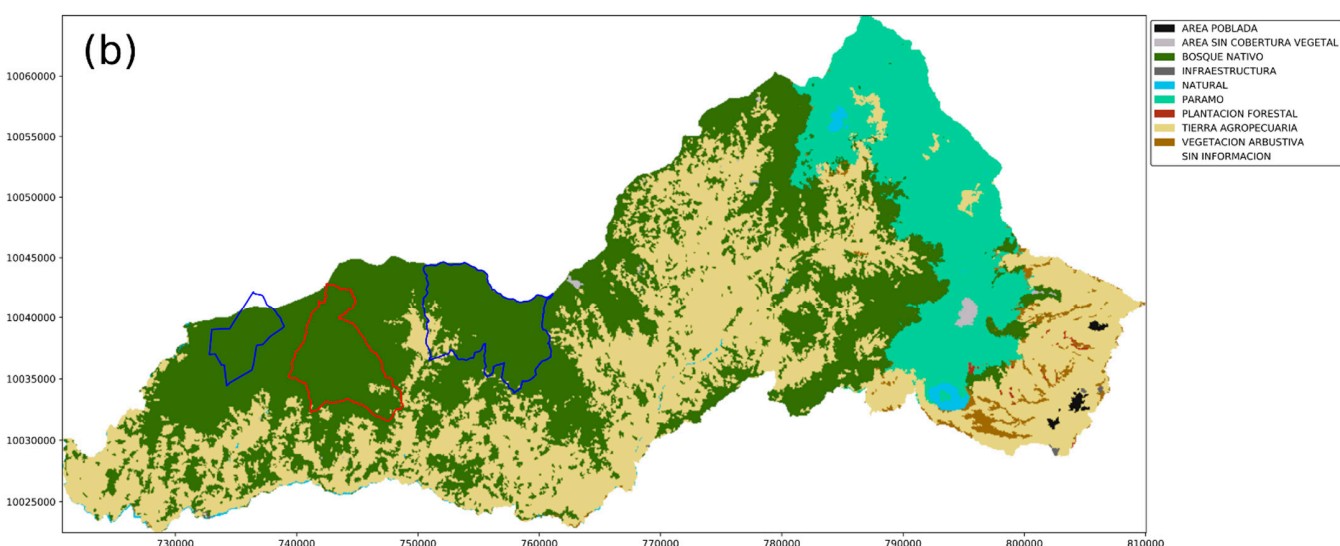

**Figure 10.** Land-cover changes in the Cotacachi canton. Reserva Los Cedros is outlined in red, and two other nearby protected forests are outlined in blue (B.P. Cebu to the west and B.P. Chontal to the east). Forest cover is shown in dark green, and agricultural land in beige. Geographic coordinate system is PSAD56/UTM zone 17S, in meters. (**a**) 1990; (**b**) 2018.

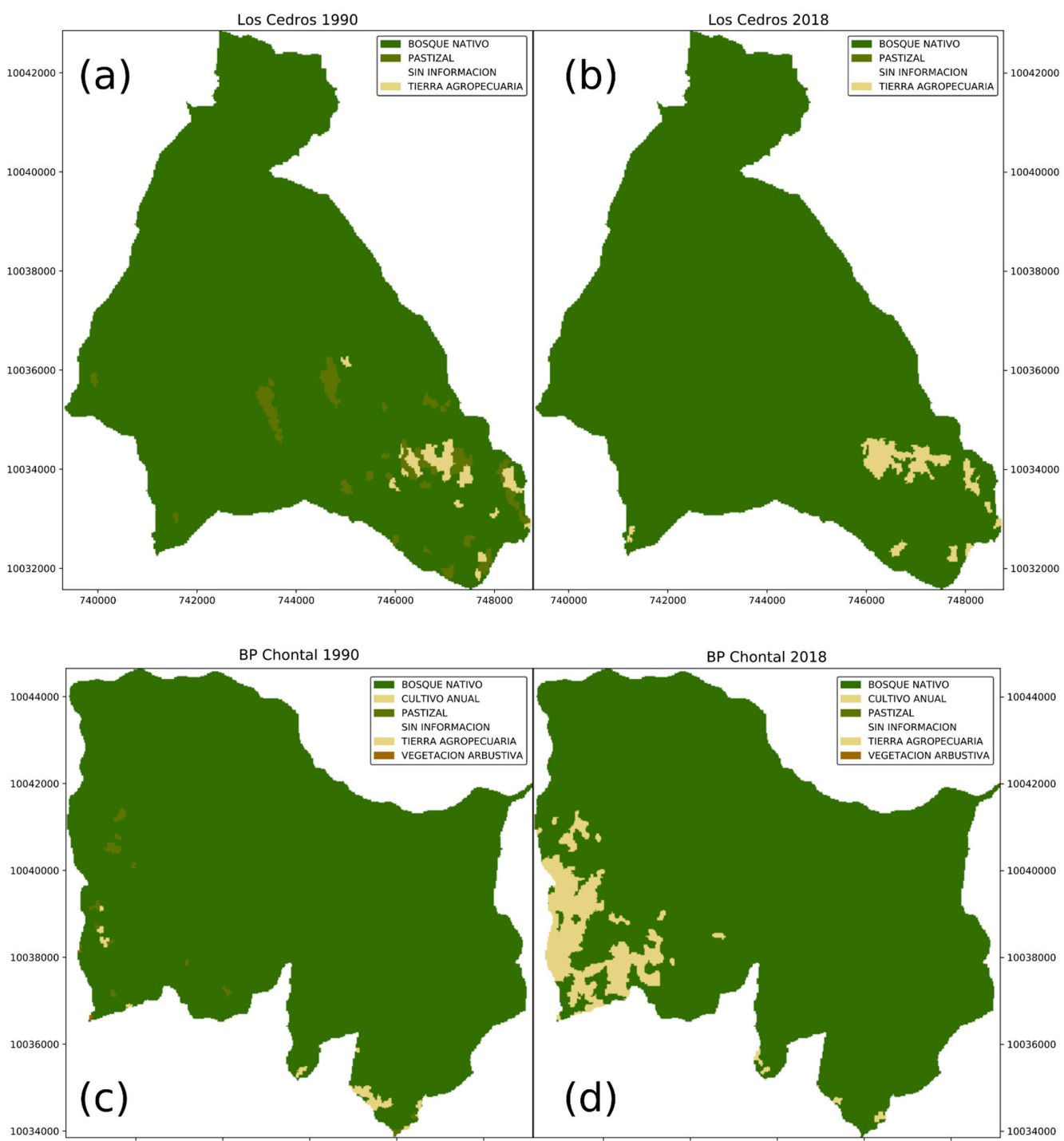

**Figure 11.** Land-cover changes in Reserva Los Cedros and in nearby protected forest B.P. Chontal. Forest cover is shown in dark green, pasture in tan-green, and agricultural land in beige. Geographic coordinate system is PSAD56/UTM zone 17S, in meters. (**a**) Los Cedros, 1990; (**b**) Los Cedros, 2018; (**c**) B.P. Chontal, 1990; (**d**) B.P. Chontal, 2018.

## 4. Discussion

### 4.1. Stable States in the Andean Cloud Forest

In this study, we examined 61 forest sites with known histories of anthropogenic disturbance or natural, gap-forming disturbance, and examined this history as a predictor of current ecological state. We could categorize the southern area of the Los Cedros reserve

into four forest types based on adult tree communities (Figure 4), and could well predict these ecological states from past land use/habitat type and elevation. Sites that had no history of anthropogenic disturbance group were readily categorized into either forest type III or IV (Figures 4 and 5). These sites were characterized by endemic, forest-dependent tree species (Table 4). The differences among these "natural" forest types III and IV were strongly predicted by elevation (Figures 6 and 7, Supplementary Figure S6), exhibiting elevation-dependent ecological zonation often observed in montane tropical forests [121,122]. Natural gap-forming disturbances did not change the species compositions of these sites to make them significantly different from other natural forest sites (Figure 5). As such, these natural forest types may represent stable equilibria with basins of attraction that give each some resilience to gap-forming disturbance (Figure 12).

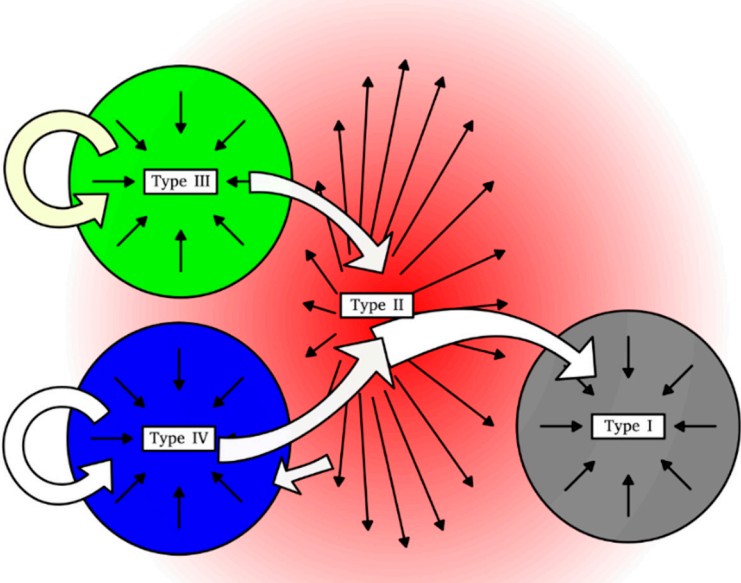

**Figure 12.** Conceptualization of ecological equilibria at Reserva Los Cedros. Arrows indicate the direction of change between basins that were hypothesized as possible from our observations. Type III and Type IV ecological states observed in our study (see Figure 5) probably represented regional historical basins of attraction that can be "knocked" into a novel forest type (type II) by intermediate agricultural disturbance, and perhaps even into a more modified type I forest by more intensive anthropogenic disturbance. Type II forests may also be capable of transition back to ecological states that resemble historical states, as was directly observed in our study at site 10.1 (see Figure 4).

There were indications that sites that had experienced intermediate disturbance (conversion to pasture followed by reforestation) had entered a less stable, more pluripotent ecological state. Nearly all high and low elevation sites that were converted to pasture prior to reforestation (land-use/habitat type "RG") then developed into a single community type, ("forest type II"). This suggested a homogenizing effect on tree communities due to this kind of disturbance and a common successional response by the forest after conversion to pasture and abandonment, regardless of elevation. In ordinations of community similarity (Figure 5, Supplementary Figure S3), type II forest sites were situated in an intermediate position between all other states. It is therefore possible that the type II forest type represents a low-sloped, convex, unstable equilibrium that can sometimes allow sites to return to a natural forest state. In one case, we observed a site that had undergone anthropogenic disturbance and subsequently developed into a natural forest type (site 10.1, circled in Figures 4 and 5). Additionally, in the case of two pastured (RG) sites, their adult communi-

ties resembled the highly affected type I forest, but their juvenile communities currently more closely resembled the "natural" type III forest (Figure 9, sites 1.2 and 1.3). In a different successional direction, several pastured (RG) sites developed into the novel Type I forest, the same forest type into which all the intensive-agriculture sites developed (Figure 5).

The existence of this indeterminate, intermediate ecological state, forest type II, supported the hypothesis that primary cloud forest has some capacity to "repair" highly modified agricultural sites, but also indicated that this is not a certain outcome. This postpasture successional trajectory appeared to be very distinct from that which followed natural, gap-forming disturbances, and may be intermediate to severe agricultural conversions (cane production followed by natural regeneration), which tended to drive sites to another state, forest type I. The adult tree communities of sites that experienced intensive agriculture all developed into forest type I, suggesting that forest type I may be another stable equilibrium or novel ecosystem type [34]. Additionally, all sites that were observed to be in a type I state were low-elevation sites that presumably would have otherwise existed in a type IV state, suggesting vulnerability of these lower-elevation forests to anthropogenic disturbance. However, all type I sites were also colocalized to the same area of the reserve (Figure 6), making it difficult to generalize this pattern to the rest of the study area.

Several forces may be at work in creating these equilibria that we observed. In type III and type IV forests ("natural forest" types), gap-forming incidents appear to keep these sites within their basins of attraction, and the breakdown of these negative feedbacks and ecological conditions that are likely responsible for shifts into forest type I equilibria.

### 4.1.1. Soil Structure

While soil structure data were not available for this survey, researchers at field sites noted that sites that had undergone intensive agricultural use ("RCA" sites) had more compacted soils. Agricultural use can incur long-lasting legacy effects on soil structural and chemical characteristics, even after abandonment and reforestation [123,124]. Deforestation has been shown to cause long-term changes in soil physical structure and microbial activity in soils, especially among plant-symbiotic microbes [125–127].

### 4.1.2. Soil Seed Bank Depletion

In addition to soil compaction, several seasons of indiscriminate grazing or corn or sugarcane culture probably greatly depleted the residual seed bank of forest plants [128,129]. When soil-seed banks have been exhausted, recolonization of sites by forest plants must occur through dispersal of seeds to the forest regeneration site.

### 4.1.3. Large Seeded Plants and Animal Dispersal

Reserva Los Cedros hosts three species of primates: the brown-headed spider monkey (*Ateles fusciceps fusciceps*), the white-headed capuchin (*Cebus capucinus*), and the mantled howler monkey (*Alouatta palliata*), in addition to numerous other frugivorous birds and mammals [52]. These animals likely play an essential role in closing gaps after disturbances. Indeed, it is likely that the differences between our results and those observed at the Maquipucuna Cloud Forest Reserve [130,131], only about 40 km away in a straight line and at similar elevations, resulted from the presence of primates at Los Cedros and their extirpation due to hunting at Maquipucuna [52]. Myster recorded little colonization of pastures, whereas ours were filled with *Cecropia*, which are commonly dispersed by both *Ateles* and *Cebus* monkeys [132,133]. In recent natural gaps ("CLB"), we observed a single indicator species: *Endlicheria* sp. (Table 4). *Endlicheria* tend to have classic bird-dispersed fruits, a small drupe with red color in the cupule [134], but some *Endolicheria* species have been suggested elsewhere to also be primate-dispersed [135]. Lower-elevation natural forests (forest type IV) were characterized by copal trees (*Protium* and *Dacryodes* spp., Table 4), well known locally as primate food sources [136]. In general, type III and IV forest types were generally characterized by plants with larger seeds or fruits, often primate- or otherwise vertebrate-dispersed, such as those in Lecythidaceae and Lauraceae, and

*Garcinia* (Table 4). This prevalence of trees with larger seeds/fruits is typical of tropical forests, where most trees produce animal-vectored seeds [137]. Absent active transport of seeds by animals, colonization of pastures by forest trees is expected to be very slow [138] and to heavily favor wind-dispersed seeds [139] or rapidly spreading cover shrubs, such as bamboo species [140]. Many ruderal species, conversely, are wind-dispersed [141] and are likely to readily colonize abandoned field sites, especially when aided by human vectors [142]. Given Los Cedros' relatively low elevation among montane tropical forests, it is not surprising that its forests are heavily populated with large-seeded species that likely rely on primate dispersal [143]. Additionally, we found four large-seeded tree species that are commonly associated with primary forests and that are animal fodder, but that were also observed as having a strong presence, even in highly disturbed sites: *Caryodaphnopsis theobromifolia* [144,145], *Clarisia biflora* [146], *Guatteria megalophylla* [147,148], and *Ficus cuatrecasana* [149] (see Table 4, indicator species for group I + IV and group II + III). *C. theobromifolia* is a valuable timber species in the area, and its presence in highly disturbed sites is welcome economically and more evidence for the high regenerative potential in these forests, perhaps due to primate dispersal. A study of forests in Peru that were protected from hunting versus those that were not only found seedlings and juveniles of Caryodaphnopsis in protected forests, and they documented more primate and mammal dispersers in the protected forests [144].

4.1.4. Relative Fluxes of Local vs. Exotic Plant Types

Most of the sites examined here were embedded in a landscape of primary forest, and may have been able to recruit seeds from forest-dependent species even if they experienced seed-bank depletion [150], while still insulated from outside seed sources. Approximately half of the type I forest sites, however, were located on the edge of Los Cedros abutting neighboring farmland, probably allowing for extensive input of small-seeded pioneer tree species such as *Saurauia* spp. The remaining type I sites were located along mule trails that supply the reserve, and also presumably were repeatedly exposed to seeds from outside locations.

The above mechanisms are primarily "gap-filling" mechanisms, and may fail in the face of extensive anthropogenic disturbances such as habitat fragmentation and loss (for example, Cramer et al. [151]). It is probably important to note how dependent these processes may be on large animal dispersers, and especially on primates, as primate populations are in decline in the region [52,152]. The importance of seed-dispersal services by large mammals and spider monkeys in particular can be appreciated by anticipating their loss: Peres et al. [153] have predicted substantial loss of forest biomass in forests if large mammal populations are reduced, especially spider monkeys and tapirs. The importance of the brown-headed spider monkey as a dispersal agent of larger seeds, often representing tree species sought after by loggers, was clearly seen in a study in Western Ecuadorian lowland forest systems [154]. A similar positive feedback may be possible with the loss of forest-dependent birds due to deforestation. Frugivorous, forest-dwelling birds provide a unique seed-dispersal service to forest trees, but are highly interdependent with their habitat; their decline in diversity has been directly linked to vegetative diversity loss and deforestation [155].

Other complex interactions may also be important: the presence of *Cecropia* spp. as an indicator species in forest types with intermediate disturbance (forest type II) and appearance less often in the novel type I forest. In this study, they were also not observed as extremely prevalent in recent natural gaps. This is possibly important, as *Cecropia* trees occupy a somewhat unique position of being an early-successional species while also producing large fruits and canopy structure characteristics useful to both primates and frugivorous birds [136,156,157]. *Cecropia* trees may attract primate dispersers and forest-dependent birds back to a disturbed site, which therefore bring with them the heavier seeds of other forest-dependent, heavy-seeded species. At Los Cedros, the smallest primate dispersers, capuchin monkeys, are regularly observed in *Cecropia* trees. *Cecropia* spp. may therefore act as another gap-repairing negative-feedback mechanism, one that plays out

in cases of forest gaps of larger size or severity than the smaller natural gaps examined here. Thus, the success or failure of *Cecropia* may play a deciding role that allows sites to return from the ecological state we observed here as forest type II (resulting from intermediate disturbance), and potentially back to primary-like forest states. Note again, however, that this mechanism would depend upon the presence of large animal vectors, especially primates.

Given the relatively short period of time since disturbance prior to surveys (13–18 years), it was also very difficult to confidently project the stability of the here-proposed candidate equilibria to larger time scales, especially in times of deep ecosystem change due to climate change. Sites were surveyed in 2005, so an additional 16 years had passed since the collection of these data. Indeed, Loughlin et al. [76] suggested that a tropical Andean cloud forest in a nearby ecoregion may have a single long-term ecological equilibrium. However, conditions in the Andes are changing dramatically from those that presumably sustained the long-term resiliency observed by Loughlin. Thus, the candidate alternative states observed here should not be disregarded, as they may mark the beginning of novel forest types in the region [34]. For the moment, visual inspection of these sites in the present day confirmed that unique tree communities continue to exist in type I forest sites compared to surrounding older forests. Updated systematic surveys are now necessary to investigate the stability of the ecosystem types suggested here.

### 4.2. Juvenile Tree Community

It is difficult to assign cause to the lack of structure observed in juvenile tree communities at Los Cedros (Figure 9). Much of the noise in these juvenile communities presumably stemmed from the incomplete filtering of young plants at each site [158,159]—the environmental pressures that have shaped the adult tree community at each site likely had yet to totally act on the juvenile trees at the time of sampling. Additionally, plots with a history of nonanthropogenic disturbances (BS and CLR plots) were not sampled for juvenile trees, meaning that we could not comment directly on the importance of environmental filtering on seedlings due to "natural" successional patterns.

However, it is also inevitable that the same global environmental changes that are acting on other forests throughout the world [27,160], including cloud forests [81], are at play in the ancient forests of Los Cedros, and are contributing to the reorganization of future tree communities, perhaps uncoupling ancient species associations. The disturbance regime under study (conversion to agriculture, followed by forest regeneration) also introduced new conditions and plant species to Los Cedros. Thus, in these juvenile trees, we may also be observing two sources of disturbance that could shift the forests of Los Cedros out of the basins of attraction of the primary forest state of very different scales: local land-use change and global climate change. In the terminology of Beisner et al. [161], the former may still be considered a state variable change from which it is sometimes within the capacity of the cloud forest to rebound and recover to a primary forest state. The latter, however, is a deep shift in the parameters of the Andean cloud forest ecosystem, which will no doubt change the shape of the possible [162–164]. It is not unreasonable to expect interactions between these two fundamental sources of ecological disturbance [26,162,165,166].

### 4.3. Beta Diversity and Spatial Heterogeneity in the Andean Cloud Forest

We examined patterns of distance decay in the tree communities of the southern area of Los Cedros. When community turnover was modeled as a function of Euclidean distance, a model in the form of an asymptote function fit well to the observed patterns of community turnover. Our asymptote model suggested turnover at a short distance, reporting that half of the maximum dissimilarity was reached in just ~150 m, and a mean Bray–Curtis dissimilarity >0.8 in comparison with distances larger than 600 m (Figure 2). When small watersheds were used as the basic spatial unit, rather than Euclidean distance, most of the decay in the tree community similarity occurred with the first crossover to a neighboring watershed (Figure 3). Following this, the mean dissimilarity among sites increased slightly

but remains uniformly high, and further comparisons were not statistically significantly different, meaning comparisons between sites five watersheds away were not on average more or less similar than comparisons of sites that were only two, three, or four watersheds apart, because so much change in community composition had already occurred just within the first watershed crossing. This was supportive of the colloquial understanding that in the Andes, each small drainage can host an almost entirely distinctive community from its neighbors.

This may also provide more insight into the processes that create the fine-scale of endemism often observed in Andean forests. We hypothesized that this fine-scale, watershed-based community turnover was due to high dispersal limitation and high microsite variability that resulted from the complex, dramatic topography of the Andes. This microsite variation was visible to some degree in our Moran's eigenvector maps and their environmental correlations (Figure 8, Supplementary Figures S9 and S10, Table 5). These MEM maps showed spatially explicit patterns of difference in tree community that were acting at very small distances, close to the scale of the microwatersheds we have delineated, and that correlated with watershed characteristics such as local elevation changes, proximity to water, and hill tops and rivers as possible dispersal barriers. These observed spatioenvironmental patterns may explain up to 19% of variance in the plant community. This fraction may represent much of the environmental filtering that is occurring within the tree community at Los Cedros. Dispersal limitation is harder to test, but the predominance of large-seeded species observed in our ancient forest sites suggested that many important tree species were dispersal-limited to highly local scales and dependent on large animal dispersers to overcome this limitation. This preponderance of heavy-seeded species also might be well approximated by a symmetric dispersal-limited neutral model, a model that can generate significant small-scale spatial patterning in communities even without considering additional microsite environmental conditions [17]. Future studies should incorporate watershed units, as they can explain more in the Andes than simple distance or elevation.

### 4.4. Conservation Value of Los Cedros

Since the sounding of an alarm by Gentry and Dodson [72,87], Myers et al. [66,167], and numerous others [78], there has been much concern about the future of the Chocó and the tropical Andean biodiversity hotspot by conservation groups. However, there is little evidence that this call for conservation has resulted in sufficient meaningful change for the region of Los Cedros—in fact, quite the opposite. Instead, during this time frame, Cotacachi Canton has lost significant forest cover (Figure 10), as has Ecuador generally [87,168,169]. In stark contrast, Los Cedros—which has onsite, conservation-oriented staff—has well withstood the traditional pressures of timbering and settlement, as evidenced by its increase in forest cover during a time of net forest loss in the Cotacachi Canton. In addition to the historical habitat conversion from timber extraction and settlement, the tropical Andean biodiversity hotspot has now found itself in a new center of metal mining exploration [52,170,171], a new and entirely different extractive pressure on the region. Los Cedros itself is targeted by a junior mining partner (cornerstoneresources.com, accessed on 31 April 2021). Los Cedros has responded with a successful legal challenge that reached Ecuador's highest courts, with implications for all Bosque Protectores in Ecuador [170]. Due to its proactive defensive legal efforts, Los Cedros' conservation effect was thus amplified even beyond the unusually effective physical protection of its forests.

Going forward, however, conservation of primary forest reserves such as Los Cedros will face novel challenges. It is widely understood that the high endemism of the Andean biodiversity hotspot makes it both a conservation priority and an especially difficult conservation challenge. It is less widely appreciated that forests, and especially primary forests, pose a major additional challenge to conservation. In most cases, we have only a crude understanding of the numerous ecological interactions required to maintain a primary ecosystem in its current levels of biodiversity and benefits for human society ("ecosystem

services"), otherwise known as the ecological complexity of an ecosystem [172,173]. Ancient ecosystems are unique in large part due to their emergent complexity, and quantifying this complexity is usually difficult, requiring composite measures of numerous indices of forest physical structure, plant community, and other ecological properties [174,175]. In complex ecosystems, the loss of a species is multiplied by disruption of its interactions with other species, such as with trophic cascades [176]. The number of possible interactions among species undergoes quadratic growth as biodiversity increases linearly, so the immense biodiversity of the tropics presents a particular challenge in understanding the local ecological complexity. Primary forest fragments are also presumably subject to the well-known vulnerabilities of "island"—or insular—habitats [177,178]; namely, the stochastic extinction of species without any nearby "mainland" to replenish populations. This is in addition to the uniquely dynamic, weather-dependent boundaries that make all cloud forests insular systems, primary or otherwise, and that may make them very vulnerable to shrinkage from climate change [72,81,164]. Thus, in insular, highly interdependent ancient ecosystems such as primary cloud forests, these stochastic species extinctions may have greater cascading impacts than in simpler ecosystems. Even if completely protected from external changes, primary forest fragments such as Los Cedros may ecologically simplify or "decay" with time if they are not sufficiently understood, buffered, and interconnected. As such, the regional resiliency noted by Loughlin et al. [76] may be subject to novel vulnerabilities that were not so urgent historically. Michaels et al. [179] have suggested that restoration efforts should first attempt to diagnose the ecological conditions and feedbacks that allow a basin of attraction for a desired stable state to persist, and also find the minimum "nucleus" area needed for those processes to proceed and grow. They and others, such as some proponents of rewilding [180], suggest seeding landscapes with species consortiums intended to recreate ecological complexity as much as possible. Los Cedros is far from needing rewilding, but conservation efforts around Los Cedros and other ancient tropical forests will be greatly enriched by such an awareness of protecting and nurturing ecological complexity, beyond simple physical protection of reserve borders.

Given the state of fragmentation and loss of forest in the Choco Andean region, and the poor understanding of its numerous pollination and diaspore dispersal networks, especially the threatened state and habitat requirements of its primate dispersers [152], Los Cedros may be at or close to threshold stable patch size for maintaining its level of biodiversity. Los Cedros may therefore face an important juncture, with a choice of growing or subsiding. Further reduction or anthropogenic disturbance of cloud forest area risks nonlinear, catastrophic changes. Additionally, the fragments of Andean primary cloud forest habitat that persist to date are rare and small enough that there is no economic justification for eliminating or heavily modifying them. Simply as a matter of scale, any economic benefit from the degradation of these forest fragments will not be large enough to justify their loss as reserves of genetic information, providers of ecosystem services, and protectors of species diversity. Other methods for poverty reduction are available [181,182]. However, great opportunity for the future exists in Los Cedros: the forest of Los Cedros need not be seen as a delicate ecological island in a storm, nor is its protection an academic or moral exercise, or a desperate rear-guard conservation strategy. It is instead useful to think in the sense of Michaels [179], supported by the long-term observations of Loughlin et al. [76], and see the Los Cedros forest as a nucleus of primary cloud forest that has prospered for thousands of years, that has weathered the recent waves of deforestation, and that now stands ready to help reseed and restore the new forests of the northern Andes.

**Supplementary Materials:** The following are available online at https://www.mdpi.com/article/10.3390/f13060875/s1, Figure S1: Sampling design, natural forest sites, Figure S2: Comparison of predictive performance of models of community turnover, Figure S3: Posterior shift for $K_m$, Figure S4: Ordination of Hierarchical clustering results (forest type), Figure S5: Predictions of current ecological state by historical land-use/habitat, Figure S6: Prediction of primary forest type by elevation, Figure S7: Current forest type, history, and elevation, Figure S8: MEM 1, Figure S9: MEM 2, Figure S10: MEM 8, Figure S11: Adult and juvenile tree communities, with adult forest

type, outlier retained, Table S1: Species lists, Table S2: Species diversity estimates for historical land-use/habitat types.

**Author Contributions:** Conceptualization, A.M., D.C.T., B.A.R. and M.P.; methodology, A.M., D.C.T. and M.P.; formal analysis, D.C.T.; investigation, A.M., R.M., A.H., W.D., M.A.C., J.D.S.L. and E.J.; resources, M.P.; data curation, A.M., B.A.R. and D.C.T.; writing—original draft preparation, D.C.T.; writing—review and editing, A.M., D.C.T., B.A.R. and M.P.; visualization, D.C.T.; supervision, M.P.; project administration, M.P. and A.M.; funding acquisition, M.P. All authors have read and agreed to the published version of the manuscript.

**Funding:** Field and herbarium work stages were funded by the Darwin Initiative under project number 14–040. Publication costs were funded by the Deutsche Forschungsgemeinschaft (DFG, German Research Foundation, 491183248) through the Open Access Publishing Fund of the University of Bayreuth.

**Institutional Review Board Statement:** Not applicable.

**Informed Consent Statement:** Not applicable.

**Data Availability Statement:** Publicly available datasets were analyzed in this study. This data can be found in the associated github repository (https://github.com/danchurch/losCedrosTrees), and all analyses are recorded as a Jupyter notebook (https://nbviewer.org/github/danchurch/losCedrosTrees/blob/master/anaData/MariscalDataExploration.ipynb).

**Acknowledgments:** The work would not have been possible without Jose DeCoux and his dedication to the conservation of Reserva Los Cedros. Several field assistants, volunteers, and guides associated with Los Cedros helped with data collection: Laurence Duvauchelle, Gila Roder, Martín Obando, Manuel Moreno, Danny Cumba, Homero Sánchez, Fausto Lomas, and Víctor Lomas. Additional help in the field and herbarium were contributed by: David Neill, Mercedes Asanza, Homero Vargas, Nelson Moyano, Gabriela Montoya, Sonia Sandoval, Iralda Yépez, Byron Nuggerud, Diana Fernández, Efraín Freire, Elsa Toapanta, Carlos Cerón, Walter Palacios, Edison Jiménez, Rosa Masaquiza, Rosa Batallas, Fernando Asanza, Marcia Peñafiel, and Laura Mariscal. Additional PrimeNet project colleagues included: Diego Tirira, Mercedes Gavilanes, María Isabel Estévez, and Xavier Cueva.

**Conflicts of Interest:** The authors declare no conflict of interest. The funders had no role in the design of the study; in the collection, analyses, or interpretation of data; in the writing of the manuscript, or in the decision to publish the results.

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
