# Peer review of "Evidence for Alternate Stable States in an Ecuadorian Andean Cloud Forest"

_forests, doi:10.3390/f13060875_

Round 1

Reviewer 1 Report

This study takes place at Reserva Los Cedros in Andean Ecuador.  It is a tropical montane cloud forest reserve where there are relatively undisturbed primary forests and forests that have histories of anthropogenic disturbance (i.e., abandoned small family farms with land used for maintaining cattle and intensively farmed sites used for agriculture). The authors did their due diligence in using hierarchical clustering of sites to delineate the different types of communities that were present at the Reserva Los Cedros.   The authors also performed an indicator species analysis and spatial analysis in these communities.  In addition to examining the adult communities, the authors also examined the juvenile communities of most of their sites.  The goal of this study was to determine how resilient communities are that have been disturbed versus those that are primary forest.  They found that primary forests are resilient, but the new juvenile community forming underneath the canopy does not resemble that of the adult tree composition. This could indicate that in the future these primary communities may look quite different. Trees that received anthropogenic disturbance may form alternate ecological states and may not return to the composition of the primary forest. The authors also use watersheds to understand changes in community biodiversity patterns as a function of topographic complexity in the tropical Andes. The authors used land cover changes from 1990 to 2018 to understand deforestation pressure in the region. The authors show that conservation efforts at Reserva Los Cedros in Andean Ecuador has managed to reverse deforestation within its boundaries.

Lines 243 How large were the subplots? I know that you answer this question later in the manuscript but as a reader, it is a question I had as I tried to visualize your design.  The size of your plots needs to be included here.  This is a simple fix.

Line 252 Why were juvenile trees in the BS plot not surveyed?  Again, the answer to this question needs to come sooner rather than later so that the reader does not question why that plot was not sampled. This is a simple fix.

 Line 334-335 I am not familiar with watershed crossings so including the link for readers is helpful for understanding how the watershed delineations were made.

 Lines 382-385 Forest types III and IV need to be defined first as “two distinct natural forest types…” If not, the reader is initially confused about why a logistic regression model is being used on these two forest types, in particular.   In your manuscript you introduce them first and define them second.  That just needs to be reversed.

Line 431 Bosque Cerrado is BC not CB.  The letters just need to be switched here.

Lines 471-674 The results are well presented and understandable to the reader.  The analyses seem appropriate for the hypothesis being addressed.

Lines 675-962 The authors do a thorough job of discussing and giving meaning to the results.  Even though the authors never examined soils, they did a good job of discussing their importance.

Reviewer 2 Report

This topic is very interesting and important to readers who are related with forest conservation under high pressure of deforestation. For benefits of readers, it would be better to revise some points from the current text.

Comments

  • Figure and Supplemental Figure do not have captions. This is a major problem.

  • The results of the spatial analysis of environmental factors in MEMs 1, 2, and 8 were analyzed, but I was unable to determine how these factors were related to the changes in forest types classification and transition. The analysis of this spatial distribution is indicated to be used to understand the spatial scale at which the tree community changes, but I could not read how it was specifically utilized in the discussion.

  • Did the RCA and RG plots begin to regenerate to forest at approximately the same time? Also, if it is agricultural land, was there any significant difference in the years of its use between the plots?

  • The impact of fire on forest succession is likely to be significant, but is there any influence of fire disturbance in the test plots?

  • Table 4: What is the significance of indicator species across the first sorting groups, Group I and IV, Group II and III?

  • In Tables 4A and 4B, the species names are confusing. For example, could it be made easier to read by separating the family name and species name with a slash?

  • In Supplemental Figure 3, the species are color-coded by forest type, but it would be better if the symbols were changed for each land use history.
